# WBA: Word Boundary Attention for Chinese Named Entity Recognition

Zhongguo Xu *

School of Computer Science and Technology, Tongji University, Shanghai, Shanghai, China

* 1910672@tongji.edu.cn

## Abstract

Chinese words often exhibit a parallel structural relationship within sentences, while individual characters are sequentially connected. To capture this structural distinction, we extract the head and tail positions of characters within words and incorporate them into a relative positional encoding scheme. Building upon this design, we introduce Word Boundary Attention (WBA), a mechanism that assigns dynamic attention weights to characters and enhances their representations with contextual information derived from the word lattice. By explicitly modeling word boundaries, WBA effectively suppresses noise, improves word recognition, and leverages richer lexicon-based context during training. Extensive experiments across multiple datasets demonstrate that WBA consistently outperforms existing approaches, achieving, for instance, a 2.51% improvement over the base model on the Weibo dataset with YJ lexicon encoding. Furthermore, visualizations of the learned attention weights reveal the interactive relationships between words and characters, providing interpretable insights into the process of word discovery. The source code of the proposed method is publicly available at https://github.com/na978292231/WBA/tree/main/WBA4NER-main.

## Introduction

Named Entity Recognition (NER) is traditionally viewed as a task involving both entity boundary detection and entity type classification. However, it is worthwhile to consider entity boundary detection as a dedicated task [1]. Identifying the start and end positions of an entity mention in text is crucial for boundary detection, independent of type classification [2]. For enumerating all possible text spans within a sentence, entity boundary representations are typically formed by concatenating start/end token representations [3], with span width captured via a 20-dimensional embedding [4] and boundary diffusion [5], consistent with prior research.

Various attention-based models have been explored for entity span detection. Yu et al. [6] employ a unified multimodal Transformer, while Zhang et al. [7] leverage an adaptive co-attention network. Additionally, boundary smoothing has been introduced as a technique that refines entity boundary detection rather than label prediction [8].

**Data availability statement:** All data underlying the results are available within the manuscript, its Supporting information files, and from publicly accessible sources. The external datasets used in this study can be accessed at the following locations: the Weibo NER dataset (https://github.com/cchennlp/weiboNER), the MSRA dataset (https://github.com/lemonhu/NER-BERT-pytorch), the OntoNotes 4.0 dataset (https://catalog.ldc.upenn.edu/LDC2011T03), and the Resume NER dataset (https://github.com/jiesutd/LatticeLSTM).

**Funding:** This work is supported by the National Natural Science Foundation of China (No. 72071145). The funders had no role in study design, data collection and analysis, decision to publish, or preparation of the manuscript.

**Competing interests:** The authors have declared that no competing interests exist.

Several studies have also explored boundary representation as a fundamental aspect of language structure in modeling [9–12].

To enhance performance, external information sources have been incorporated, including lexicons [13], glyph features [14,15], syntactic structures [16], and semantic representations [17]. More recently, pre-trained language models such as BERT [18–20], GPT [21–28] have further boosted performance.

In the context of Chinese NER, purely character-based methods [29–31] often fail to fully utilize word-level information. Chinese NER poses additional challenges due to its character-level structure (Fig 1a), which typically requires word segmentation and assumes equal spacing between adjacent characters. Since named entities in Chinese are composed of words, attending to word boundaries within a sentence is crucial.

Actually, Chinese words are logically parallel in a sentence. In Chinese tasks, NLP tools build related lexicon to match the terms [32]. For instance, in Fig 1b, "南京市" (*Nanjing City*) forms a word-level structure, while "南京" (*Nanjing*) represents a subword-level structure. Both structures can function as named entities with a nested representation in NLP tasks. At the subword level, the distance between "南" (*South*) and "京" (*Capital*) is shorter than that between "京" (*Capital*) and "市" (*City*). At the word level, "京" (*Capital*) and "市" (*City*) are closer together than "市" (*City*) and "长" (*Long*). These observations indicate that word boundaries play a crucial role in defining entity boundaries. Therefore, an effective approach to Chinese NER should incorporate both word embeddings and word position information.

Different segmentation styles correspond to different downstream tasks [33–35]. For instance, in Named Entity Recognition (NER), the boundary of an entity must be defined as "长江大桥" (*Yangtze River Bridge*), whereas in Neural Machine Translation (NMT), segmentation can separate "长江" (*Yangtze River*) and "大桥" (*Bridge*).

To address these segmentation challenges, we introduce a word-level structure compilation approach for NER. We derive word start and end positions from segment labels in datasets or established tools such as Jieba, LTP, and CoreNLP. We further enhance attention models by incorporating word position information, leading to WBA, which utilizes a relative Transformer architecture. The main contributions include:

- To enhance word-level representation, we leverage segmentation tags to identify the head and tail positions of words, enabling knowledge discovery of word structures. Based on these extracted word boundary points, we propose a simple yet effective model built upon a relative Transformer architecture that explicitly attends to word boundaries within a sentence.
- We integrate word boundaries into character representations, introducing WBA for Chinese NER and helping the model automatically suppress noise and identify correct words.
- We evaluate WBA on multiple benchmark datasets, demonstrating its superiority over state-of-the-art approaches. Our proposed method is transferable to different sequence-labeling architectures and can be easily adapted for other NLP tasks.

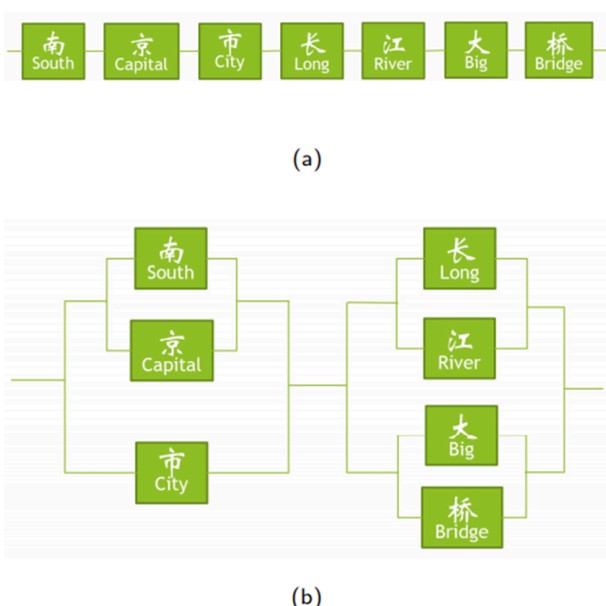

**Fig 1**. **Chinese input with different structures.** (a) Character-level structure. (b) Word-level structure.

## Related work

Integrating word information into character-based Named Entity Recognition (NER) systems has been a major research focus in recent years, often referred to as word augmentation. Due to the scarcity of annotated NER data, pre-trained language models such as BERT tend to underperform on certain NER tasks. This limitation is particularly evident in Chinese NER, where word-enhancement methods have been shown to outperform or closely match BERT's performance, underscoring their necessity. A variety of approaches have been proposed to effectively incorporate word information into character-based models.

**Lexicon- and lattice-based approaches for Chinese NER.** Chinese NER is challenging due to the lack of explicit word delimiters. Prior approaches have integrated lexicon information to alleviate segmentation ambiguity, using lexicon-enhanced embeddings [13], lattice structures [36], or hybrid character–word models [37]. Adaptive word embedding methods such as WC-LSTM [38], NMDM [39], and CAN-NER [40] incorporate vocabulary information in a model-independent, transferable manner. Other techniques, including SoftLexicon [41] and ConvTransformer [42], exploit lattice structures and convolutional layers to capture both word-level and subword-level contexts. SubRegWeigh [43] further introduces effective and efficient annotation reweighting with subword regularization. While lattice-based models allow multiple word candidates to be represented simultaneously, they often introduce noise and increase computational complexity. To address this, recent studies propose dynamic weighting of lexicon information and refined character–word interactions, including Lattice LSTM [13], LR-CNN [44], LGN [45], and CGN [46], which explicitly model word structures. Our method differs by mining explicit head–tail boundary points of words and incorporating them into attention, preserving the benefits of lexicon-aware modeling while reducing noise from irrelevant candidates.

**Transformer-based models and positional encoding.** The Transformer architecture has become the foundation of modern sequence modeling, with self-attention mechanisms enabling efficient context capture across tokens [47]. Numerous variants extend the Transformer by modifying positional encodings to better capture structural or relational information [48,49]. Relative positional encodings improve the modeling of long-range dependencies, while boundary-aware mechanisms have been applied to parsing and segmentation tasks [50]. Transformer-based models such as Lattice

Transformer [51], FLAT [37], NFLAT [52], and pre-trained models like ZEN [53], LEBERT [36], and BERT-wwm [54] incorporate lexicon-aware self-attention to boost performance. Our work builds on this line of research by explicitly encoding word boundary positions, which differentiates it from standard self-attention and most positional encoding methods that treat words or characters uniformly.

**Noise suppression and interpretability in NER.** Beyond structural modeling, researchers have investigated noise suppression in NER datasets through label denoising [55], confidence-based reweighting [56], and distributional analysis [57]. Visualization and interpretability methods have also been developed to analyze attention mechanisms and their relationship to linguistic structures [58]. Our work contributes to this line of research by demonstrating that explicit boundary modeling not only improves performance but also yields interpretable attention maps that reveal interactions between words and characters in Chinese sentences.

Our study is particularly inspired by FLAT and NFLAT, which significantly advance Chinese NER by incorporating lexicon information through lattice structures with dual positional encodings in relative attention. However, these models do not explicitly capture word boundaries, which can lead to errors and inefficiencies in character-level enhancements. To overcome this limitation, we propose a word boundary enhancement method that leverages relative attention to explicitly attend to word boundaries, thereby achieving superior performance compared to existing approaches.

## Background

NFLAT introduces the InterFormer method, which significantly simplifies the attention matrix used in FLAT. Additionally, the context feature encoding from TENER enables the model to effectively attend to character-level positions. Together, these two enhancements contribute to achieving relatively high performance in Chinese NER.

As illustrated in Fig 2a, NFLAT streamlines FLAT's approach by computing attention scores solely for 'character-word' interactions. Additionally, it introduces the '< non_word >' tag for single characters that do not match the lexicon or correspond to punctuation marks, setting their attention values to zero. This prevents unnecessary attention allocation to non-relevant words. The model then employs character-based TENER to compute 'character-character' attention scores using relative position encoding, incorporating both orientation and distance awareness. By decoupling lexicon fusion from context feature encoding, NFLAT achieves improved accuracy and efficiency over FLAT.

As shown in Fig 2b, NFLAT mitigates issues related to blurred word boundaries and the absence of word semantics. However, for richer and more precise word representations, we argue that computing 'character-character' interactions within word boundary attention is essential. We use a novel word boundary guided relative position encoding, rather than lattice-based fusion.

To address this, we introduce an additional step between word-level embedding and the TENER encoder attention. Our proposed Word Boundary Attention (WBA) explicitly attends to word boundaries, further enhancing both accuracy and efficiency beyond NFLAT.

## Methods

For Chinese Named Entity Recognition (NER), WBA consists of four main stages. In the first stage, word-level embeddings are applied to integrate semantic information from words. The second stage introduces WordFormer, which captures word boundary information within the sentence. In the third stage, a Transformer Encoder is employed to encode contextual information enriched with lexicon features. Finally, a Conditional Random Field (CRF) [59] is used as the decoder to predict the sequence labels. The WordFormer module is a key contribution of this work. An overview of the WBA architecture is illustrated in Fig 3.

### Word-level embedding

In this work, we implement this layer with different models, e.g., base model (NFLAT [52]), pre-trained model ( BERT-wwm [54] and ZEN [53]).

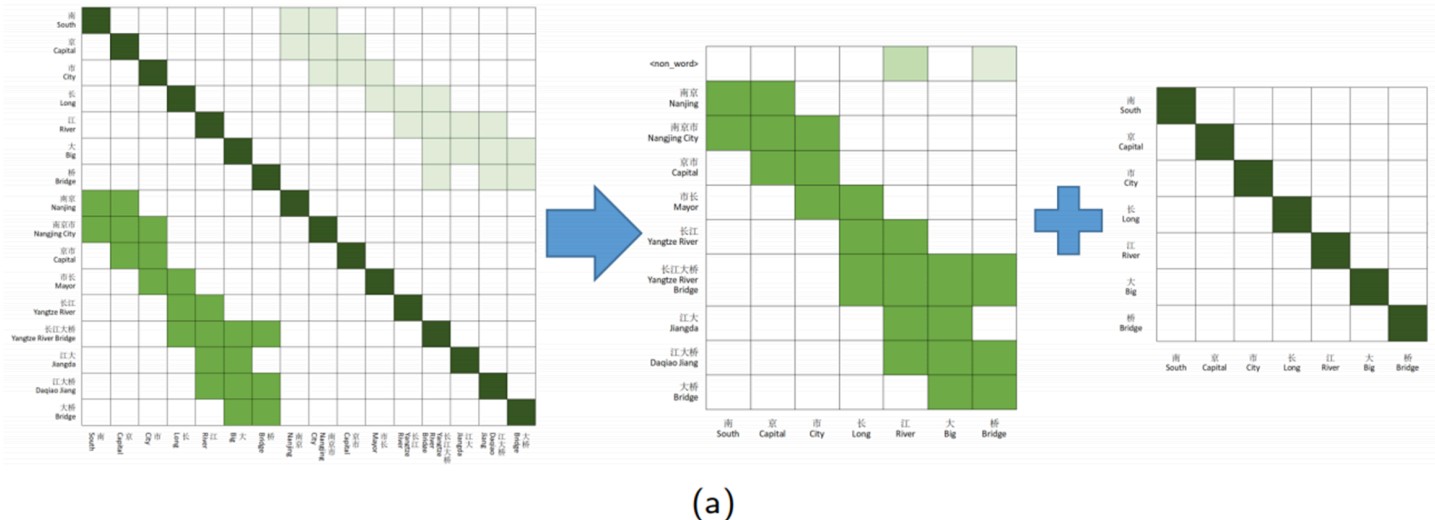

(a)

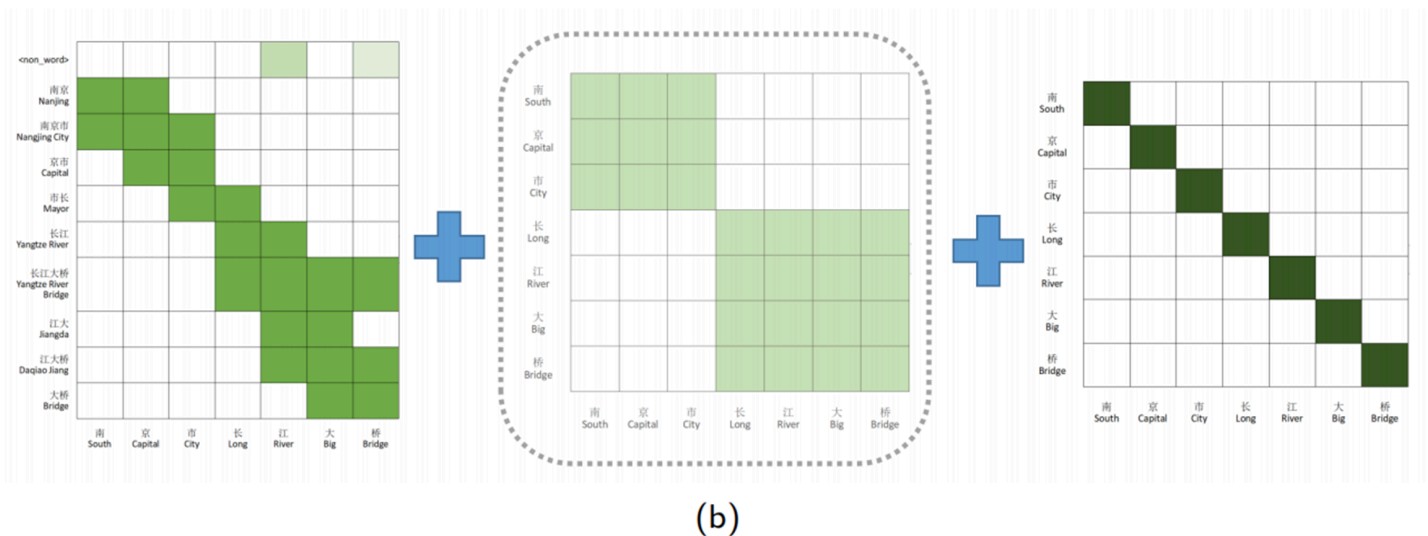

(b)

**Fig 2. NFLAT and its attention matrix.** (a) The self-attention matrix of NFLAT from FLAT (b) The self-attention matrix of NFLAT to WBA.

For base model, we use unigram and bigram embeddings in static character embedding layer. Each character $c_t$ is embedded as $\boldsymbol{x}_{c_t}$. Each word $w_t$ is embedded as $\boldsymbol{x}_{w_t}$. The character embedding encoder represents the sentence based on the character sequence $c_1, c_2, \ldots, c_n$ and the word embedding encoder represents the sentence based on word sequence $w_1, w_2, \ldots, w_m$.

$$\boldsymbol{x}_{c_t} = \mathbf{e}^c(c_t), \tag{1}$$

where $\mathbf{e}^c$ denotes a character embedding lookup table.

$$\boldsymbol{x}_{w_t} = \mathbf{e}^w(w_t), \tag{2}$$

where $\mathbf{e}^w$ denotes a word embedding lookup table.

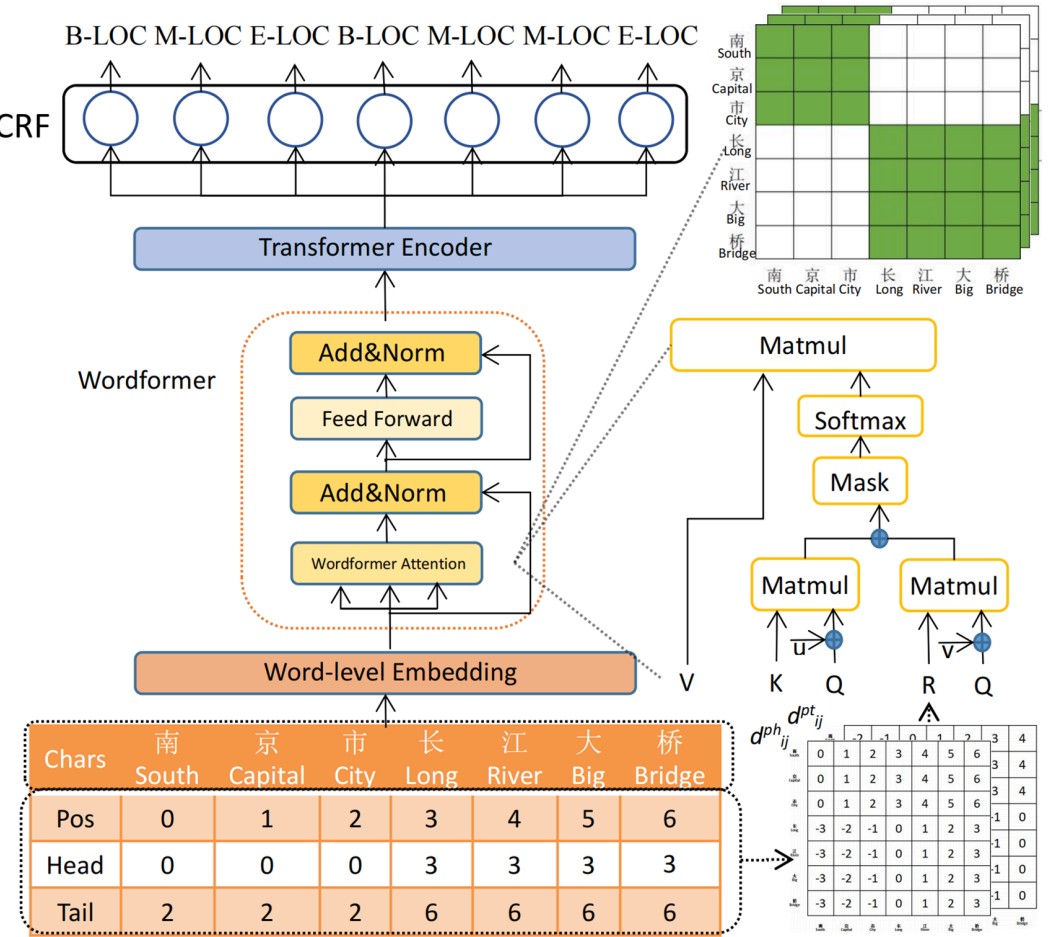

**Fig 3**. The overall architecture of WBA.

Word-level embedding aims to construct a non-flat-lattice and jointly model two sequences of characters and words with different lengths. It enables the sequence of characters to fuse word boundaries and semantic information. This part is inspired by NFLAT. The inputs $Q$, $K$, and $V$ are derived through linear transformations of character and word feature embeddings:

$$[Q, K, V] = \left[X^c W_q, X^w W_k, X^w W_v\right],\qquad(3)$$

where $X^c = \{x_{c_1}, x_{c_2}, ..., x_{c_n}\}$ and $X^w = \{x_{w_1}, x_{w_2}, ..., x_{w_m}\}$ denote the token embeddings for the character and word sequences, respectively. Each of the projection matrices $W_q$, $W_k$, and $W_v$ is a learnable parameter.

To incorporate lexicon information, we employ an inter-attention mechanism that fuses character and word representations and generates the hidden states $h_{c_i}$:

$$\mathrm{InterAtt}\,(A, V) = \mathrm{softmax}\,(\mathrm{mask}(A))\,V,\qquad(4)$$

where *mask()* is the inter-attention score mask of characters and words, which is 2-dimensional for a single batch and 3-dimensional for a multi-batch. It is used to fill empty positions in the sequence with the value of $10^{-15}$, so that the

attention weights of these positions are close to 0 when *softmax()* normalization. The input attention is calculated in a similar way to NFLAT, using two relative positions.

For pre-trained models, the character sequence are put into the pre-trained modeling layer to obtain hidden states $\boldsymbol{h}_{c_i}$, which models the word-level information between characters.

For example, in Fig 3, the character sequence ['南' (*South*), '京' (*Capital*), '市' (*City*), '长' (*Long*), '江' (*River*), '大' (*Big*), '桥' (*Bridge*)] is encoded to generate character hidden states $\boldsymbol{h}_{c_i}$. Meanwhile, the lexicon sequence ['< *non_word* >', '南京' (*Nanjing*), ..., '大桥' (*Bridge*)] is encoded to obtain lexicon embeddings $\boldsymbol{x}_{w_t}$, which serve as word-level embeddings in the base model.

## WordFormer attention

After word-level embedding, each character has the information of lattice. We use WordFormer attention to enable the sequence of characters to fuse word boundaries in a sentence, which is absent in vanilla Transformers. Unlike the standard Transformer, which applies self-attention uniformly across tokens, WBA introduces explicit head and tail nodes to represent word boundaries. This modification allows the model to capture word-level structural information, which is critical for Chinese where word segmentation is ambiguous. Furthermore, WBA restricts and reweights attention based on boundary positions, thereby reducing noise from irrelevant characters. These innovations distinguish WBA from standard attention and contribute directly to improved NER performance.

The input $\boldsymbol{Q}$, $\boldsymbol{K}$, $\boldsymbol{V}$ are obtained by the linear transformation of rich chars feature hidden states from word-level embedding encode:

$$[\boldsymbol{Q}, \boldsymbol{K}, \boldsymbol{V}] = \left[\boldsymbol{h}^c \boldsymbol{W}_q, \boldsymbol{h}^c \boldsymbol{W}_k, \boldsymbol{h}^c \boldsymbol{W}_v\right], \tag{5}$$

where the hidden states of character sequences, $\boldsymbol{h}^c = \{\boldsymbol{h}_{c_1}, \boldsymbol{h}_{c_2}, ..., \boldsymbol{h}_{c_n}\}$. Each of the projection matrices $\boldsymbol{W}_q$, $\boldsymbol{W}_k$, and $\boldsymbol{W}_v$ is a learnable parameter.

We use word segment tags or tools to split the sentence into words. Then we estimate word position probabilities while handling ambiguous word boundaries to reduce systematic errors and biases into the model. It begins by enumerating all possible word boundary candidates. Next, it applies character-lexicon attention, using word match positions to compute the probability of each character appearing at the beginning (B), middle (M), end (E), or as a single-character word (S). Finally, the framework integrates masking and class attention mechanisms to generate informative position features for characters, leveraging both character and lexicon contexts.

For word-level structure, we set head and tail nodes of words. Each character $c_i$ has position $p_i$, head $h_i$ and tail $t_i$ nodes. Each word is separated in the sentence. We consider each character in the position of its word, using two relative distances like NFLAT:

$$d_{ij}^{ph} = p_i - h_j, \tag{6}$$

$$d_{ij}^{pt} = p_i - t_j, \tag{7}$$

where $d_{ij}^{ph}$ denotes the distance between position of $p_i$ and head of $h_j$, and $d_{ij}^{pt}$ denotes the distance between position of $p_i$ and tail of $t_j$. Fig 3 shows examples of $d_{ij}^{ph}$ and $d_{ij}^{ph}$ in a more intuitive form.

The final relative position encoding of words is a simple non-linear transformation of the two distances:

$$R_{ij} = \text{ReLU}(W_r(\mathbf{p}_{d_{ij}^{ph}} \oplus \mathbf{p}_{d_{ij}^{pt}})), \tag{8}$$

where $W_r$ is a learnable parameter, $\oplus$ denotes the concatenation operator, and $\mathbf{p}_d$ is calculated as in Transformer [60],

$$\mathbf{p}_d^{(2k)} = \sin\left(d/10000^{2k/d_{model}}\right), \tag{9}$$

$$\mathbf{p}_d^{(2k+1)} = \cos\left(d/10000^{2k/d_{model}}\right), \tag{10}$$

where $d$ is $d_{ij}^{ph}$, $d_{ij}^{pt}$, $k$ denotes the index of dimension of position encoding and $d_{model}$ is the hidden size.

We then apply a variant of self-attention to incorporate relative word position encoding, as illustrated in Fig 3.

$$\boldsymbol{A}_{ij} = (\boldsymbol{Q}_i + \boldsymbol{u})^\top \boldsymbol{K}_j + (\boldsymbol{Q}_i + \boldsymbol{v})^\top \boldsymbol{R}_{ij}^*, \tag{11}$$

where $1 \leq i \leq n$, $1 \leq j \leq m$. The $\boldsymbol{u}$, $\boldsymbol{v}$ are learnable parameters. Eq (11) is from Dai et al. [61].

The input attention is computed similarly to NFLAT, utilizing two relative positions as defined in Eq (11) and Eq (4). As illustrated in Fig 3, the attention heads highlight the word boundary between "南京市" (*Nanjing City*) and "长江大桥" (*Yangtze River Bridge*). The characters ['南' (*South*), '京' (*Capital*), '市' (*City*)] share the same head and tail positions within the word segment "南京市" (*Nanjing City*) and primarily focus on each other while paying comparatively less attention to the other characters in the sequence.

### Transformer encoder

After the WordFormer module, character features are enriched with both lexicon information and word boundary signals. Next, a Transformer encoder is employed to capture contextual dependencies. This component is inspired by the work of Yan et al. [62], which demonstrated that unscaled self-attention and relative position encoding with directional information are particularly effective for NER tasks. Finally, a Conditional Random Field (CRF) decoder is applied to predict the most likely label sequence by maximizing the overall sequence scores.

## Experiments

We evaluate the proposed WBA using standard metrics—F1 score (F1), precision (P), and recall (R)—and compare its performance with several character-word hybrid models. Additionally, we conduct a visual analysis of the word boundary attention weights to verify the effectiveness of the proposed mechanism. Finally, we present a comparison of results with and without the use of pre-trained language models.

### Experimental settings

**Data.** The model is evaluated on four Chinese NER datasets, including Weibo [63,64], Resume [13], OntoNotes 4.0 [65] and MSRA [66]. Weibo is a social media domain dataset, which is drawn from Sina Weibo, while OntoNotes 4.0 and MSRA datasets are in the news domain. Resume dataset consists of resumes of senior executives, which is annotated by Zhang et al. [13]. Table 1 shows the statistical information of these datasets.

**Embedding.** For base model, we download the specified pre-trained unigram and bigram embeddings for Chinese task. We conduct experiments on YJ [67] and LS [68] lexicons. Table 2 shows the statistics of the external lexicons used in this paper, mainly including their total words, single-character words, two-character words, three-character words, and words with more characters. Also, we list the dimensions of the word vectors in lexicon. Then we use relative attention incorporate lexicon in character embedding. For the pre-trained model, we use Chinese-BERT-wwm [54] and ZEN for character embedding.

**Table 1**. Statistics of the benchmarking datasets.

| Datasets | Items | Train | Dev | Test |
|---|---|---|---|---|
| Weibo | Sentences | 1.35k | 0.27k | 0.27k |
| | Entities | 1.89k | 0.39k | 0.42k |
| Resume | Sentences | 3.8k | 0.46k | 0.48k |
| | Entities | 1.34k | 0.16k | 0.16k |
| MSRA | Sentences | 46.4k | - | 4.4k |
| | Entities | 74.8k | - | 6.2k |
| OntoNotes 4.0 | Sentences | 15.7k | 4.3k | 4.3k |
| | Entities | 13.4k | 6.95k | 7.7k |

**Table 2**. Statistics of the lexicons. The *Single*, *Two*, and *Three* represent the number of characters in the word, respectively. The *Other* means the number of words with more characters. The *Dimension* is the dimension of the word embedding.

| Lexicons | Total | Single | Two | Three | Other | Dimension |
|---|---|---|---|---|---|---|
| YJ | 704.4k | 5.7k | 291.5k | 278.1k | 129.1k | 50d |
| LS | 1292.6k | 18.5k | 347.7k | 415.6k | 511.3k | 300d |

**Hyper-parameter settings.** For base model, this paper uses only one layer of word-level embedding, one layer of Word-Former encoder and one layer of Transformer encoder to handle Chinese NER. We adopt similar settings as NFLAT [52]. Most implementation details include character and word embedding sizes of 50, a dropout rate of 0.2, and a batch size of 10. Additionally, the hidden size is set to 192 with 12 heads for small datasets Weibo and Resume, and 256 with 8 heads for larger datasets OntoNotes and MSRA. The initial learning rate is set to 0.003 for Weibo and 0.002 for the remaining three datasets with SGD step rule. We train each model on training sets with 100 epochs totally.

For pre-trained model, most implementation details followed those of BERT-NER, including character and word embedding sizes with 786, dropout with 0.15, learning rate with 2e-5 in Adam step rule, batch size with 16. We train each model on training sets with 100 epochs totally.

## Experimental results

We conduct experiments on four datasets to further evaluate the effectiveness of our model in combination with a pre-trained model. The experiments were run on a physical machine equipped with CentOS Linux 7.9.2009 (Core), three Intel(R) Xeon(R) Gold 6226R CPUs, and two NVIDIA TITAN RTX GPUs.

Table 3 presents the results on the Weibo, Resume, OntoNotes, and MSRA datasets, comparing our approach against state-of-the-art statistical models that leverage rich word lexicon features, including Lattice-LSTM [13], WC-LSTM [38], LR-CNN [44], LGN [45], PLTE [69], SoftLexicon [41], FLAT [37], and NFLAT [52]. We use NFLAT as the baseline model. For pre-trained models, we compare with BERT-wwm [54] and ZEN [53] across all four datasets.

Our results show that NFLAT+WBA with the YJ lexicon improves upon NFLAT. Specifically, the overall F1 score on Weibo increases by 2.51%, as the dataset's lower baseline F1 leaves more room for improvement. The Weibo NER dataset, sourced from the social media platform Sina Weibo, contains informal and colloquial language, leading to blurred word boundaries. By incorporating attention mechanisms that explicitly capture word boundaries between characters, our approach proves especially beneficial for Chinese NER.

On the Resume dataset, the F1 score improves by 0.71%, while OntoNotes 4.0 sees a 0.22% increase. Both datasets contain structural information and shallow semantics, making character boundaries more easily identifiable. For the MSRA dataset, our method yields a smaller improvement of 0.12%, possibly due to differences in word segmentation, such as the phrase "中外" (*at home and abroad*) being split into "中" (*Chinese*, labeled as "S-NS") and "外" (*abroad*, labeled as "O"), which may hinder entity recognition.

**Table 3. A comparison with other lexical enhancement methods.** YJ and LS lexicon is used here. **Bold** indicates leading all baselines with $p < 0.05$.

| Models | Weibo | | | Resume | | | Ontonotes 4.0 | | | MSRA | | |
|---|---|---|---|---|---|---|---|---|---|---|---|---|
| | NE | NM | Overall | P | R | F1 | P | R | F1 | P | R | F1 |
| Lattice-LSTM | 53.04 | 62.25 | 58.79 | 94.81 | 94.11 | 94.46 | 76.35 | 71.56 | 73.88 | 93.57 | 92.79 | 93.18 |
| WC-LSTM | 53.19 | 67.41 | 59.84 | 95.27 | 95.15 | 95.21 | 76.09 | 72.85 | 74.43 | 94.58 | 92.91 | 93.74 |
| LR-CNN | 57.14 | 66.67 | 59.92 | 95.37 | 94.84 | 95.11 | 76.40 | 72.60 | 74.45 | 94.50 | 92.93 | 93.71 |
| LGN | 55.34 | 64.98 | 60.21 | 95.28 | 95.46 | 95.37 | 76.13 | 73.68 | 74.89 | 94.19 | 92.73 | 93.46 |
| PLTE | 53.55 | 64.90 | 59.76 | 95.34 | 95.46 | 95.40 | 76.78 | 72.54 | 74.60 | 94.25 | 92.30 | 93.26 |
| SoftLexicon | 59.08 | 62.22 | 61.42 | 95.71 | 95.77 | 95.74 | 77.13 | 75.22 | 76.16 | 94.73 | 93.40 | 94.06 |
| TENER | - | - | 58.82 | 94.79 | 94.97 | 94.88 | 75.97 | 77.29 | 76.63 | 92.97 | 91.96 | 92.46 |
| FLAT(YJ) | - | - | 60.32 | 94.98 | 95.21 | 95.10 | 76.75 | 77.93 | 77.34 | 94.43 | 93.85 | 94.14 |
| NFLAT(YJ) | 59.10 | 63.76 | 61.22 | 95.55 | 94.85 | 95.20 | **75.69** | 79.09 | 77.35 | 94.76 | 93.53 | 94.14 |
| NFLAT(YJ)+WBA | **59.76** | **65.10** | **63.73** | **96.06** | **95.77** | **95.91** | 75.67 | **79.58** | **77.57** | **94.86** | **93.67** | **94.26** |
| NFLAT(LS) | 59.54 | 64.82 | 63.13 | 94.40 | 95.66 | 95.02 | 73.68 | 77.55 | 75.56 | 93.69 | 92.83 | 93.26 |
| NFLAT(LS)+WBA | **59.62** | **64.95** | **63.43** | **94.84** | **95.79** | **95.31** | **74.08** | **79.02** | **76.47** | **95.36** | **94.10** | **94.73** |
| BERT-wwm | 71.64 | 59.63 | 70.66 | 96.54 | 95.95 | 96.25 | 78.47 | 80.54 | 79.49 | 94.79 | 94.97 | 94.88 |
| BERT-wwm+WBA | **72.39** | **70.77** | **71.26** | **96.64** | **96.73** | **96.68** | **78.51** | **81.09** | **79.78** | **95.21** | **95.03** | **95.12** |
| ZEN | 64.83 | 67.65 | 66.71 | 95.34 | 95.46 | 95.40 | 78.07 | 80.04 | 79.03 | 95.39 | 95.07 | 95.25 |
| ZEN+WBA | **65.28** | **68.21** | **67.65** | **95.42** | **96.27** | **95.85** | **78.11** | **80.19** | **79.18** | **95.41** | **95.23** | **95.32** |

Overall, NFLAT+WBA outperforms all other methods, including FLAT, across the four datasets. Without additional data augmentation or pre-trained language models, NFLAT+WBA achieves state-of-the-art performance on Weibo, Resume, and OntoNotes 4.0, particularly excelling in smaller datasets.

Furthermore, we evaluate NFLAT+WBA using larger lexicons (LS) and compare its performance with other models. As lexicon size increases, NFLAT+WBA improves, demonstrating that a larger and richer lexicon benefits WBA.

Additionally, WordFormer can be extracted from WBA as a standalone module, and WBA can seamlessly integrate pre-trained models into the embedding layer. As shown in Table 3, BERT-wwm+WBA and ZEN+WBA further enhances the performance of the pre-trained BERT-wwm and ZEN model respectively.

## Convergent speed comparison

We compare the convergence speed of NFLAT+WBA and NFLAT with the YJ lexicon across four datasets. Fig 4 presents the test F1 scores per epoch over 100 epochs. The results show that NFLAT+WBA consistently outperforms NFLAT in convergence speed across all four datasets. Notably, the improvement on the Weibo dataset is particularly significant, demonstrating the efficiency of our method in handling complex and informal text.

## Complexity analysis

WBA consists of three main components: (1) inter-attention between characters and words with a complexity of $\mathcal{O}(nmd)$, (2) WordFormer attention between characters with complexity $\mathcal{O}(n^2d)$, and (3) self-attention over the contextual sequence with complexity $\mathcal{O}(n^2d)$. Thus, the overall computational complexity of WBA is $\mathcal{O}((2n+m)nd)$. The corresponding space complexities are $\mathcal{O}(nm)$ for inter-attention, $\mathcal{O}(n^2)$ for WordFormer attention, and $\mathcal{O}(n^2)$ for self-attention.

We argue that WBA is particularly well-suited for Chinese NER, as demonstrated by its performance in terms of model size, time cost, memory usage, and inference time, summarized in Table 4. The reported time cost corresponds to the average per epoch on the Weibo dataset, and the model size reflects the hyperparameter configuration used for the same dataset. The memory usage is measured on the Weibo test set with a batch size of one, and the inference time represents the total accumulated duration of all test samples during evaluation on Weibo test set and its extended version.

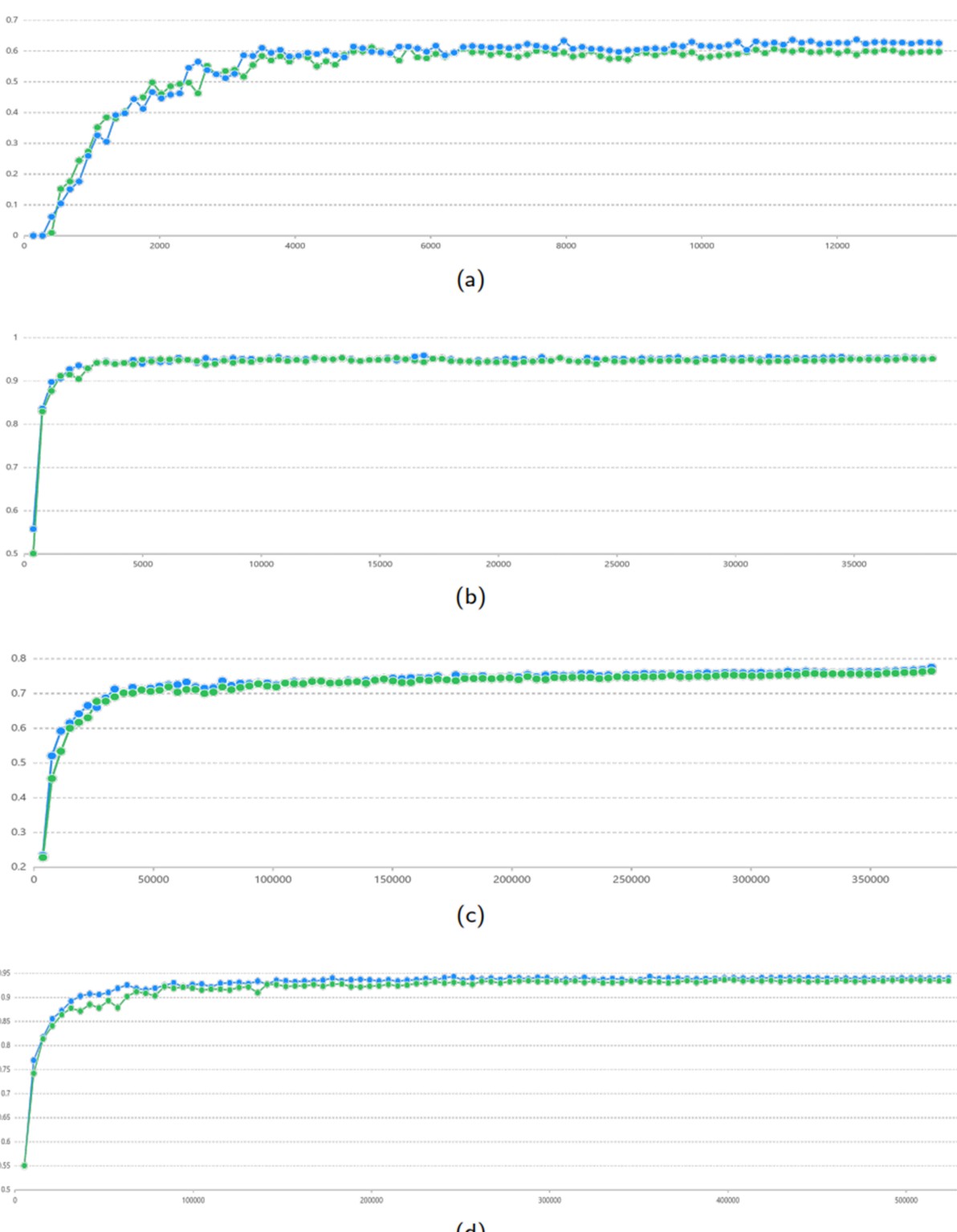

**Fig 4**. **Convergent speed comparison between NFLAT+WBA and NFLAT.** Blue points denote NFLAT+WBA and green points denote NFLAT, respectively. The x-axis and y-axis denotes test F1 and steps, respectively. (a) Weibo comparison. (b) Resume comparison. (c) OntoNotes comparison. (d) MSRA comparison.

**Table 4**. Comparison of model size, time cost, memory usage and inference time.

| Models | #Parameters(M) | Time cost(s) | Memory Usage (MB) | Inference Time (s) |
|---|---|---|---|---|
| TENER | 0.27 | 4.66 | 9.89 | 3.75/12.37 |
| FLAT(YJ) | 0.59 | 24.24 | 13.97 | 3.52/12.96 |
| NFLAT(YJ) | 0.86 | 18.45 | 14.89 | 3.83/12.53 |
| NFLAT(YJ)+WBA | 1.45 | 19.52 | 17.87 | 4.02/13.03 |

From the comparison, TENER exhibits the smallest parameter size (0.27M) and the highest computational efficiency, with a training time of only 4.66s, memory usage of 9.89MB, and inference time of 3.75s/12.37s. However, due to its lack of lexical-level interactions, it struggles to capture complex word boundary features inherent in Chinese text. In contrast, FLAT(YJ) introduces a lexicon-lattice structure, which significantly enhances its ability to incorporate external word information. As a result, its parameter size increases to 0.59M, training time extends to 24.24s, and memory usage rises to 13.97MB. Despite the higher computational cost, its inference time remains relatively low (3.52s/12.96s), indicating that FLAT maintains strong decoding efficiency even with more complex structural modeling.

NFLAT(YJ) optimizes the integration between characters and words through a non-lattice structure, effectively reducing redundant path computations. Its training time decreases to 18.45s, memory usage is 14.89MB, parameter count reaches 0.86M, and inference time is 3.83s/12.53s. These results demonstrate that NFLAT achieves a better balance between model complexity and computational efficiency.

Building upon this, NFLAT(YJ)+WBA incorporates the word boundary attention mechanism, enabling the model to adaptively capture implicit boundary information. Although its parameter size increases to 1.45M and memory usage slightly rises to 17.87MB, its training time (19.52s) and inference time (4.02s/13.03s) remain comparable to NFLAT(YJ). This indicates that WBA enhances boundary awareness and semantic discrimination without significantly increasing computational overhead.

This conclusion is further supported by the results shown in Figs 5 and 6. Fig 5 is based on samples from the Weibo test set, while Fig 6 presents an extended version of the same dataset. Together, they illustrate the inference time trends of TENER (yellow), FLAT (blue), NFLAT (green), and WBA (red) models across different sentence lengths. Overall, the inference time of all four models increases linearly with sentence length, indicating that their computational complexity is primarily driven by input length rather than nonlinear operations within the model structure.

In the short-to-medium length range (10–150 tokens), the differences among the models are more pronounced. TENER, due to its simple architecture, consistently achieves the lowest latency across all ranges; FLAT and NFLAT show comparable performance, with NFLAT performing slightly better for medium-length sentences; NFLAT+WBA shows slightly higher inference time than the others but maintains a stable trend without exponential growth, suggesting that incorporating word boundary information does not introduce significant computational overhead.

When sentence length extends to a longer range (up to 800 tokens), all four models exhibit an almost perfectly linear relationship between inference time and input length. Although all models demonstrate stable scalability for long sequences, the fitted curves reveal that NFLAT+WBA incurs slightly higher computational cost than FLAT and NFLAT, while TENER continues to deliver the best efficiency. This finding indicates that WBA, while enhancing contextual modeling capability, maintains near-linear inference complexity, effectively balancing efficiency and representational power—making it a practical and efficient choice for long-text named entity recognition tasks.

Fig 7 presents the GPU memory usage of TENER, FLAT, NFLAT, and NFLAT+WBA during training on the extended Weibo test set. By setting the training batch size to 1, we obtain the comparison of memory consumption across different sentence lengths. As the sentence length increases, all models show a clear upward trend in memory usage. Among them, TENER consistently maintains the lowest memory consumption, reflecting its lightweight architecture. NFLAT requires more memory than TENER but significantly less than FLAT and NFLAT+WBA, especially for longer

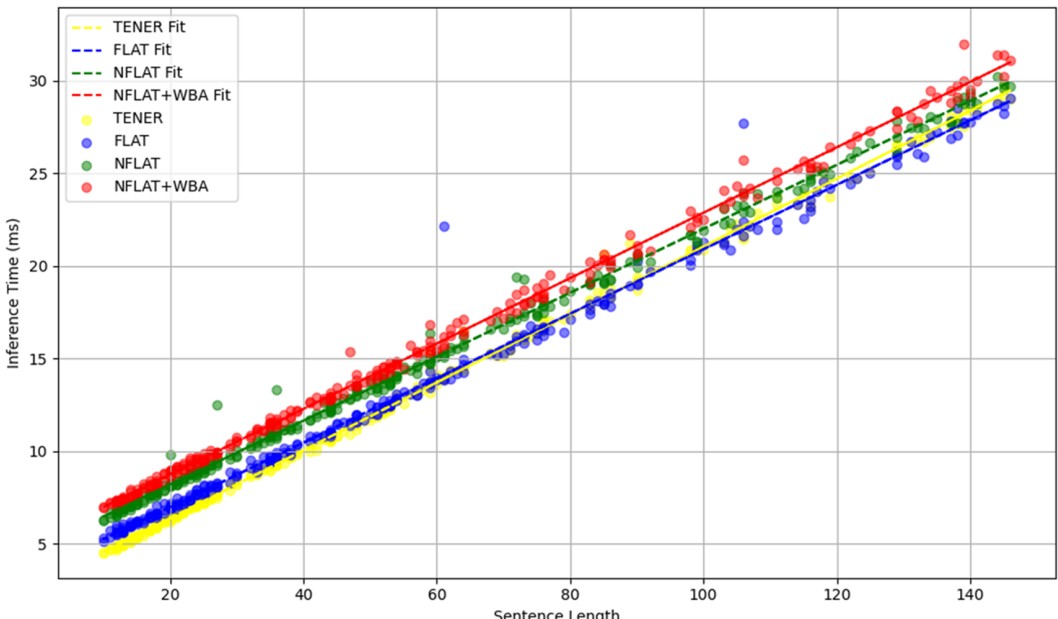

**Fig 5**. A comparison between TENER, FLAT, NFLAT and NFLAT+WBA in terms of inference time, evaluated at different sentence lengths on Weibo test set.

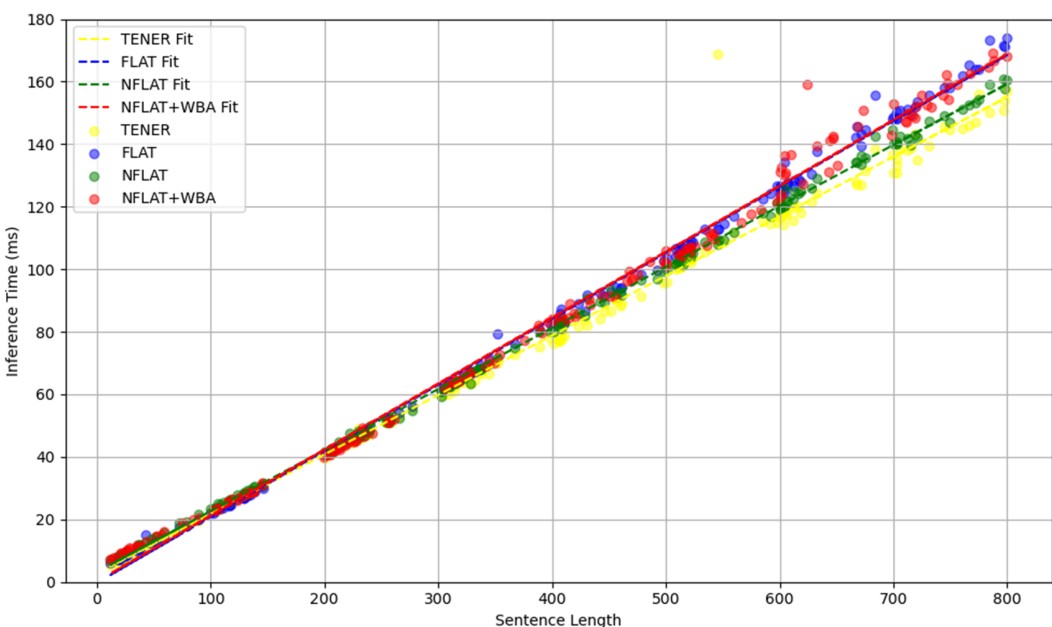

**Fig 6**. A comparison of TENER, FLAT, NFLAT, and NFLAT+WBA in terms of inference time, evaluated across different sentence lengths on the extended version of the Weibo test set.

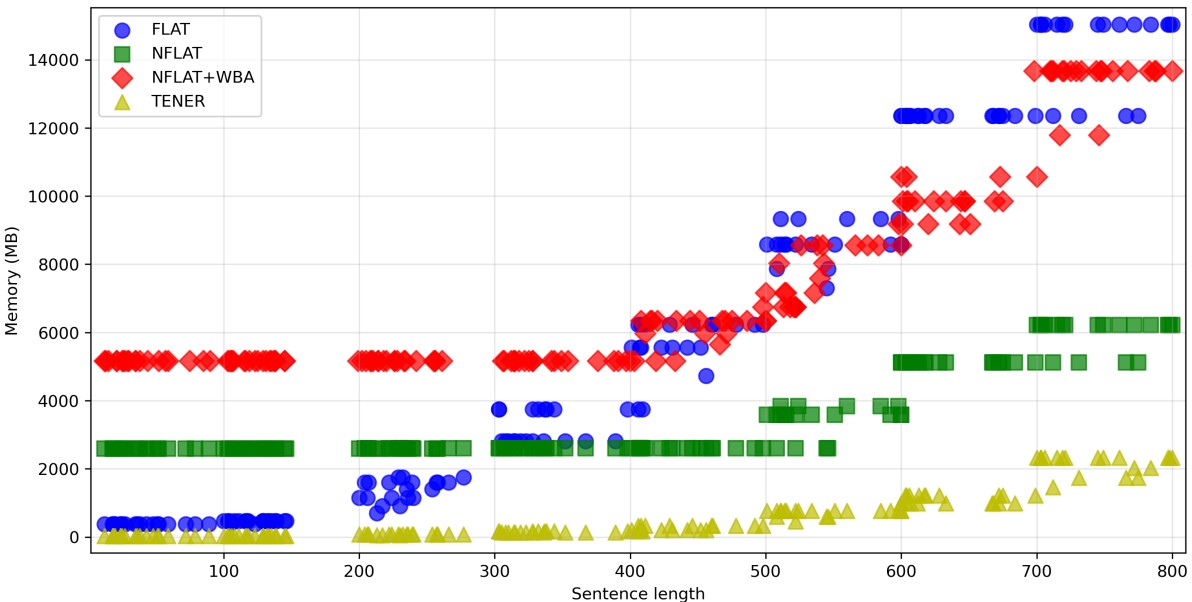

**Fig 7**. **Comparison of memory consumption during training for TENER, FLAT, NFLAT, and NFLAT+WBA, conducted on the extended Weibo test set with varying sentence lengths.**

sentences. FLAT shows slightly higher memory usage than TENER when the sentence length is below 200. As the sentence length increases, the corresponding lexicon length also grows, leading to a rapid rise in memory consumption. When the sentence length exceeds 300, FLAT surpasses NFLAT in memory usage, and when it exceeds 500, it also surpasses NFLAT+WBA. Due to the introduction of the Word Boundary Attention (WBA) mechanism, NFLAT+WBA processes additional word boundary nodes, resulting in the highest memory consumption when the sentence length is below 500. Overall, these results indicate that WBA introduces additional computational overhead, yet it still demonstrates better scalability than FLAT when handling long sequences.

By jointly analyzing the table and figure, we can conclude that WBA effectively enhances boundary detection and semantic modeling while keeping computational complexity nearly unchanged. Compared with FLAT, NFLAT+WBA achieves a better balance across model size, training cost, and inference efficiency, demonstrating both structural compactness and practical feasibility. These results provide strong evidence that WBA is an efficient and robust enhancement mechanism for Chinese NER, maintaining linear scalability while significantly improving modeling performance.

## Cross-dataset validation

To demonstrate the generalizability of the proposed method,we conduct additional experiments using the CoNLL-2003 [70], Wiki-ja, and Konec dataset.

In the processing of the CoNLL-2003 dataset, each line of data consists of a token and its corresponding entity label. The entity labels use the BIO annotation scheme, where B- represents the beginning of an entity, I- represents the inside of an entity, and O represents non-entity tokens. To add head and tail nodes for each line, we process it based on the syntactic component label in the third column. Each B- labeled token is considered the head of a new entity block, and subsequent I- tokens share the same tail node. Tokens labeled as O are treated as individual blocks with the same head and tail node number. When an empty line is encountered, the numbering is reset to 0, and a new block numbering starts. In the experimental setup, the pre-trained embedding is GloVe 100d, the learning rate is set to 9e-4 and the batch size is set to 100 based on the parameters of the Weibo dataset.

In the processing of the Wiki-ja dataset, we first perform Japanese tokenization using tools such as MeCab to segment the text into words. If the word is not a number, it will be split into characters and re-labeled to fit the different entity block requirements. For multi-character words like "東京大学" (*Tokyo University*), the entire word is treated as a block, and the head and tail nodes for each character in the word are assigned the same value. The first character of the entity is labeled with B-, and the remaining characters are labeled with I-, while non-entity characters are labeled with O. When an empty line is encountered, the numbering is reset, and the head and tail nodes are adjusted to ensure that each sentence is independently numbered. In the experimental setup, the pre-trained embedding is FastText 300d, the learning rate is set to 5e-3 based on the parameters of the Weibo dataset.

In the processing of the Konec dataset, head and tail nodes are assigned as follows: when encountering a line with an empty character column, we skip the line and continue with the next numbering. When a line contains punctuation marks, it is treated as an independent block. Additionally, when encountering an empty line, the numbering is reset. In the experimental setup, the pre-trained embedding is FastText 300d, the learning rate is set to 3e-3 and the batch size is set to 50 based on the parameters of the Weibo dataset.

The results are shown in Table 5. On the English dataset CoNLL-2003, the introduction of WBA significantly improved the overall model performance. For example, in the case of TENER, the F1 score increased from 83.31 to 85.12, achieving a gain of +1.81. This demonstrates that word boundary augmentation effectively enhances the model's ability to capture entity boundaries. Since English words are naturally separated by spaces and exhibit stable morphological structures, WBA strengthens word-level information modeling without introducing additional noise, allowing the model to perform more precisely in boundary recognition and entity classification. Moreover, WBA further improves the model's recall (R), indicating that it not only captures entity boundaries more accurately but also reduces omissions, thereby enhancing the model's overall generalization across sentences. This result verifies that WBA provides a stable and transferable improvement in languages with clear morphological rules.

On the Wiki-ja (Japanese) dataset, the performance variation among models presents a more complex pattern. First, the TENER model shows a substantial improvement after applying WBA—the F1 score rises from 68.72 to 78.02, a gain of +9.30—indicating that word boundary augmentation plays a strong compensatory role in character-level Transformer frameworks. Because Japanese text lacks explicit word delimiters, word boundaries must be inferred through particles, verb endings, and semantic cues, making it difficult for traditional character-level models to accurately capture entity boundaries. The additional boundary signals introduced by WBA effectively guide the model to learn word-level structural patterns, thereby improving both precision and recall in entity recognition.

However, when transitioning from TENER to NFLAT, overall performance declines. NFLAT's F1 score drops from 73.43 to 71.75, and even with WBA, it fails to surpass the performance of TENER+WBA. This downward trend reveals two underlying causes. First, NFLAT is designed to integrate multi-granularity lexical information through a non-lattice structure. In Japanese, this mechanism can introduce granularity conflicts—the model simultaneously learns representations at the character, word-piece, and subword levels, causing competition among boundary signals and reducing feature consistency. Second, WBA may generate redundant boundary information within the NFLAT framework, overlapping with its internal multi-granularity attention mechanism and reducing training stability. Furthermore, the morphological complexity and boundary ambiguity of Japanese exacerbate these issues, making the augmented

**Table 5**. Cross-dataset validation.

| Models | CoNLL-2003 | | | Wiki-ja | | | Konec | | |
|---|---|---|---|---|---|---|---|---|---|
| | P | R | F1 | P | R | F1 | P | R | F1 |
| TENER | 85.46 | 81.27 | 83.31 | 69.65 | 67.81 | 68.72 | 70.31 | 67.84 | 69.05 |
| TENER+WBA | **87.58** | **82.79** | **85.12** | **79.14** | **76.93** | **78.02** | 79.82 | 74.90 | 77.25 |
| NFLAT | - | - | - | 75.37 | 71.59 | 73.43 | 82.61 | 80.27 | 81.42 |
| NFLAT+WBA | - | - | - | 73.74 | 69.89 | 71.75 | **83.90** | **81.54** | **82.71** |

boundaries semantically unreliable and introducing boundary noise, which ultimately leads to performance degradation. In summary, WBA exhibits a phenomenon of "strong enhancement for character-level models but limited compatibility with multi-granularity models" in Japanese tasks. This suggests that the effectiveness of WBA depends not only on the predictability of linguistic boundaries but also on the model's capacity to absorb and utilize word-level information. Future work may focus on language-adaptive boundary control strategies or granularity-level constraints to improve stability and generalization in morphologically complex languages such as Japanese.

On the Konec (Korean) dataset, WBA also yields notable performance gains. The F1 score of TENER increases from 69.05 to 77.25, a gain of +8.20, while NFLAT's performance improves from 81.42 to 82.71, showing stable results. Although Korean uses a phonetic writing system, its morphological structure is regular and its word boundaries are relatively well-defined. Therefore, WBA can guide the model to learn clearer word-boundary representations without introducing noise, strengthening the identification of entity boundaries. Especially in the recognition of long-span or compound entities, the boundary cues from WBA help the model distinguish entity from non-entity spans, significantly improving both recall and overall F1. This indicates that in languages with relatively consistent boundary rules, WBA effectively enhances cross-lingual transferability and generalization.

In summary, WBA demonstrates distinct language-dependent characteristics across datasets: it consistently improves performance on CoNLL-2003 (English) and Konec (Korean), but shows model-dependent variation on Wiki-ja (Japanese). Specifically, WBA provides strong enhancement for character-level architectures such as TENER, while in models that already incorporate multi-granularity lexical relationships (e.g., NFLAT), it may introduce redundancy and granularity conflicts, leading to decreased performance. These findings suggest that the cross-lingual generalization effect of WBA is influenced by both linguistic boundary properties and model architecture. Future research could explore language-adaptive and hierarchical integration mechanisms to further improve stability and robustness in morphologically complex languages.

## Discussion

### Ablation study

To investigate the contribution of each component of our method, we conduct ablation experiments with YJ lexicon on all four datasets, as shown in Table 6.

In the w/o Word-level Embedding experiment, we remove the word-level embedding from NFLAT+WBA. The performance drop across all four datasets highlights the importance of lexicon fusion and contextual feature encoding. However, on Weibo, the F1 score remains 1.16% higher than that of NFLAT, suggesting that WordFormer provides stronger benefits than word-level embedding in small datasets. Moreover, the consistent outperformance over TENER indicates that WordFormer effectively captures word boundaries for improved Chinese NER.

**Table 6**. Ablation experimental results.

| Models | Weibo | Resume | Ontonotes | MSRA |
|---|---|---|---|---|
| TENER | 58.17 | 95.00 | 72.43 | 92.74 |
| FLAT | 60.32 | 95.45 | 76.45 | 94.12 |
| NFLAT | 61.22 | 95.20 | 77.35 | 94.14 |
| NFLAT+WBA | **63.73** | **95.91** | **77.57** | **94.26** |
| w/o Word-level Embedding | 62.38 | 95.13 | 76.83 | 93.62 |
| w/o Transformer Encoder | 61.44 | 95.02 | 76.67 | 93.84 |
| w/o Word-level Embedding&Transformer Encoder | 61.09 | 94.89 | 76.10 | 92.61 |
| w/o WordFormer Attention | 61.20 | 95.19 | 77.30 | 94.11 |
| w/o WordFormer Attention&Transformer Encoder | 60.91 | 94.36 | 76.08 | 92.29 |
| w/o WordFormer Attention &Word-level Embedding | 58.02 | 94.20 | 71.42 | 91.98 |

In the w/o Transformer Encoder experiment, we remove the Transformer encoder from NFLAT+WBA. Compared to the word-level embedding ablation, the larger performance degradation across datasets shows that character relative position modeling is more critical than lexicon fusion. Still, the F1 score on Weibo is 0.22% higher than NFLAT, again demonstrating WordFormer's effectiveness in small datasets.

In the w/o Word-level Embedding & Transformer Encoder experiment, both components are removed. The further performance decline across datasets confirms the complementary roles of lexicon fusion and character position modeling. Nevertheless, the F1 score on Weibo is 0.77% higher than FLAT, showing that WordFormer surpasses FLAT attention on small datasets.

In the w/o WordFormer Attention experiment, we ablate WordFormer and retain NFLAT with self-attention only. The performance drop across datasets emphasizes the significance of explicit word boundary modeling. Despite this, the F1 score on Weibo is still 0.92% higher than FLAT, suggesting that NFLAT inter-attention remains stronger than FLAT attention in small datasets.

In the w/o WordFormer Attention & Transformer Encoder experiment, we remove WordFormer and rely on inter-attention with self-attention. The resulting degradation underscores the combined importance of word boundaries and character relative position modeling. Yet, the F1 score on Weibo is still 0.59% higher than FLAT, again validating the strength of NFLAT inter-attention.

Finally, in the w/o WordFormer Attention & Word-level Embedding experiment, WordFormer is removed while keeping self-attention and TENER. The performance drop further demonstrates the necessity of explicit boundary modeling together with lexicon fusion.

Overall, the ablation results demonstrate that WordFormer plays a more critical role than both lexicon fusion and the Transformer encoder, particularly in low-resource datasets such as Weibo. While lexicon fusion and relative position modeling contribute significantly to performance, WordFormer consistently outperforms these components when isolated, underscoring its effectiveness in capturing word boundaries. These findings highlight WordFormer as the primary driver of improvement in Chinese NER, offering both robustness and efficiency across diverse datasets.

### Few shots study

The model's robustness warrants further examination. To strengthen the validity of our findings under low-resource conditions, we employ few-shot learning approaches using limited dataset samples. Specifically, we simulate few-shot scenarios by randomly selecting subsets of varying sizes from the training sets of Weibo and Resume datasets, ensuring representation of all label categories. The sampling sizes (N) are set to 250, 500, and 1000 samples. Detailed sampling statistics for each label type, along with their mapped Chinese equivalents, are provided in accordance with the FE-CFNER framework [71].

The overall results are summarized in Table 7. Our proposed model consistently outperforms the baseline across evaluated datasets. Notably, the marginal improvements in the average F1-score decline as the parameter N increases. This observation indicates that our model (WBA) demonstrates superior performance in few-shot learning settings, particularly when N=250, where significant performance gains are achieved.

**Table 7**. **Results on few-shot datasets from Weibo and Resume (%). Bold** indicates leading all baselines with *p* < 0.05.

| Models | Weibo | | | Resume | | |
|---|---|---|---|---|---|---|
| | N=250 | N=500 | N=1000 | N=250 | N=500 | N=1000 |
| TENER | 38.87 | 42.08 | 50.48 | 87.52 | 89.59 | 91.82 |
| FLAT(YJ) | 38.01 | 47.06 | 54.47 | 87.83 | 90.73 | 92.48 |
| NFLAT(YJ) | 38.23 | 48.86 | 54.69 | 88.91 | 90.27 | 92.11 |
| NFLAT(YJ)+WBA | **39.76** | **49.10** | **54.73** | **90.06** | **90.77** | **92.61** |

## Analysis of Word Boundary Attention

The proposed word boundary attention (WBA) module can be understood as an interactive attention mechanism. By leveraging WBA, we design a relative position encoding strategy that enables the model to capture potential relationships between tokens in sequences with different positions. The primary goal of this work is to establish connections among character positions, heads, and tails, allowing the character sequence to effectively integrate both boundary and semantic information of words.

**Self interpretability.** As shown in Fig 8, we present a heatmap of the attention weights and corresponding entropy values generated by the word attention module. The visualization clearly highlights word boundaries between characters. Single-character words such as "谁" (*who*), "才" (*just*), "是" (*is*), and "自" (*self*) exhibit higher self-attention weights

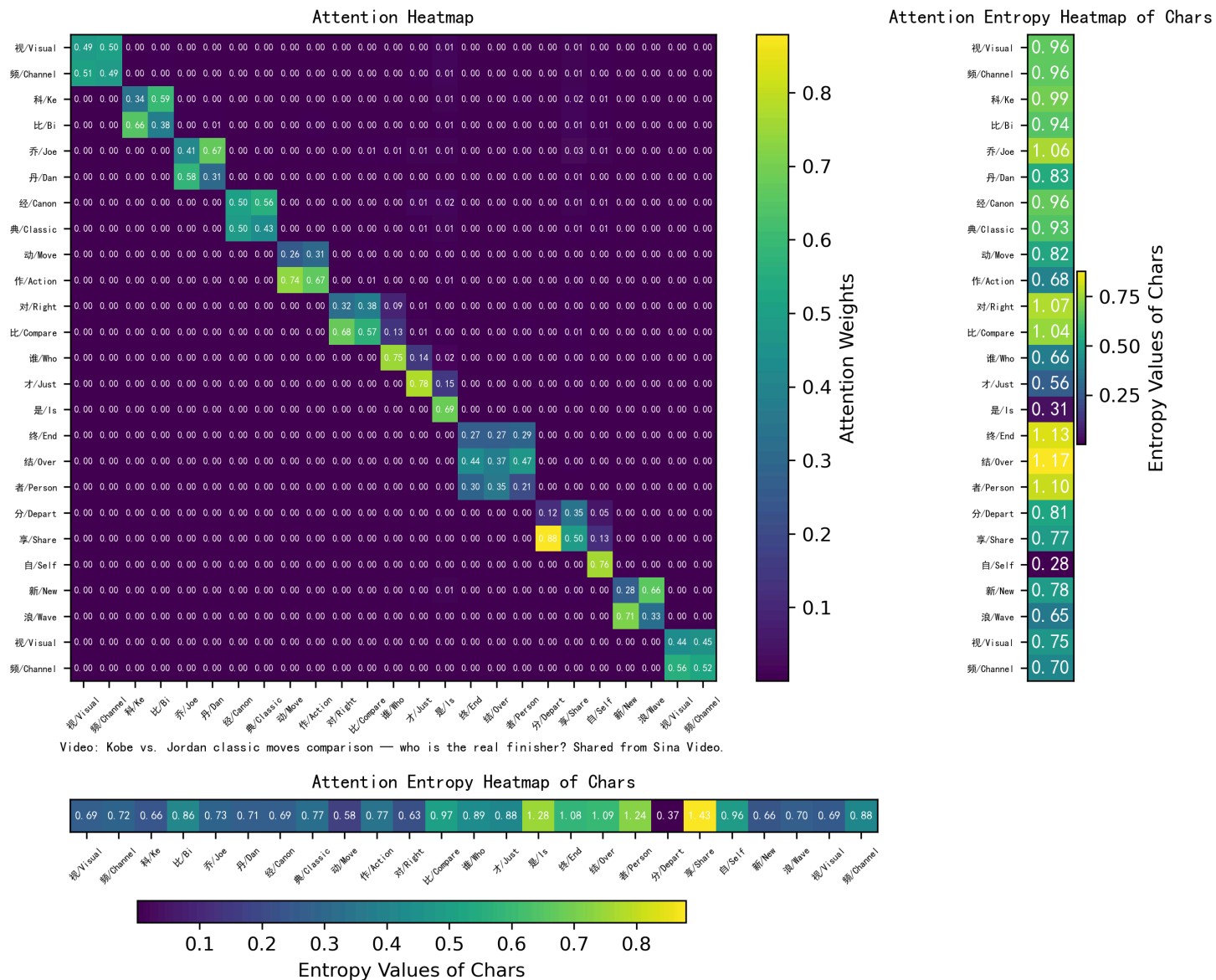

**Fig 8. Visualization of the proposed word boundary attention module.**

(0.75, 0.78, 0.69, and 0.76, respectively), while characters within multi-character words receive comparatively lower attention weights. The entropy values of single-character words are lower, indicating that their attention distributions are more concentrated. In contrast, longer words distribute attention more evenly across their constituent characters, which results in lower attention weights assigned to each character. For instance, in the word "终结者" (*terminator*, composed of "终" (*end*), "结" (*tie/over*), and "者" (*person*)), each character receives relatively modest attention from the others. Specifically, "终" (*end*) receives attention weights of 0.27, 0.27, and 0.29 from "终," "结," and "者," while attracting minimal attention from other characters. The entropy values of longer words are higher than those of shorter words, suggesting that their attention distributions are more dispersed. These findings demonstrate that the word attention mechanism is capable of automatically identifying meaningful words and appropriately allocating attention across their constituent characters.

**Comparison interpretability.**

We further analyze the behavior of different word boundary attention heads in our ablation study, as illustrated in Fig 9. In Fig 9a, where WordFormer is placed between the word-level embedding and the Transformer encoder, the model primarily attends to the first character of each word. In Fig 9b, where WordFormer follows the word-level embedding but excludes the Transformer encoder, the attention shifts toward the last character, and interestingly, characters such as "新" (*new*) and "浪" (*wave*) only attend to each other. In Fig 9c, where WordFormer precedes the Transformer encoder without word-level embedding, the last characters of words and single-character words receive noticeably lower attention weights compared to others. These observations provide strong evidence for the interpretability of the proposed method. Specifically, the word boundary attention module effectively captures boundary relationships between Chinese characters and integrates lexical information, thereby enhancing the performance of Chinese NER.

**Weights distribution comparison.** This study compares weights distributions across four Chinese NER datasets. As shown in Fig 10, statistical characterization and distribution visualization reveal long-tail patterns. The datasets exhibit similar deviation characteristics with $p < 0.05$. These findings demonstrate the effectiveness of combining statistical and visual analyses for cross-dataset quality assessment, providing quantitative support for noise suppression.

## Why Word Boundaries more informative

Different word boundaries will make different word types. Named Entity Recognition (NER) is traditionally viewed as a task involving both entity boundary detection and entity type classification. The main linguistic features are boundaries and types. The first step is boundary detection and the second step is type classification. For example, in text

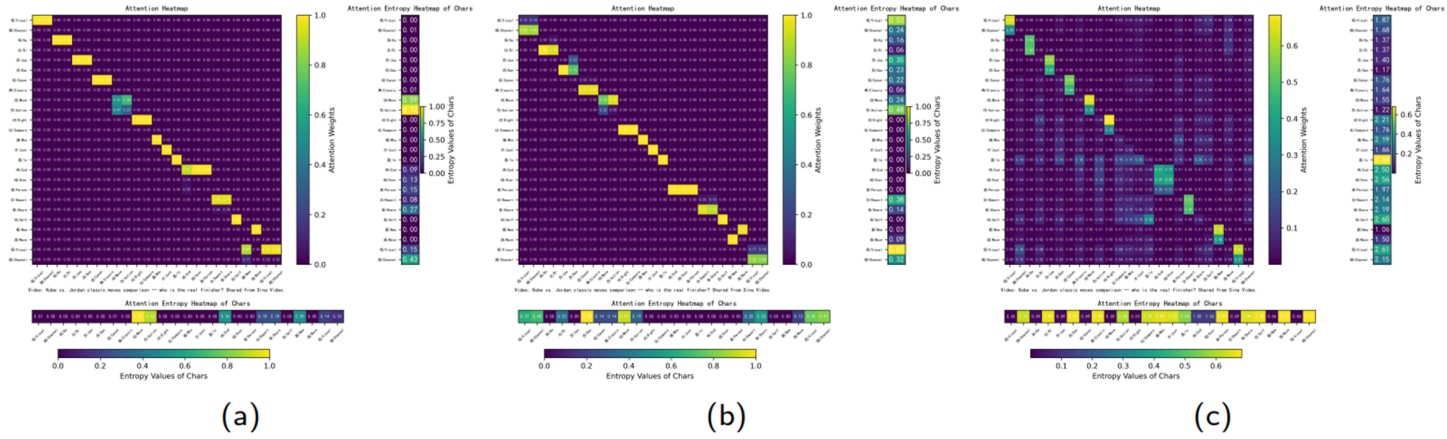

(a)                          (b)                          (c)

**Fig 9**. **A comparison among WordFormer, word-level embedding and Transformer encoder.** (a) Interpretability with word-level embedding and Transformer encoder. (b) Interpretability with word-level embedding. (c) Interpretability with Transformer encoder.

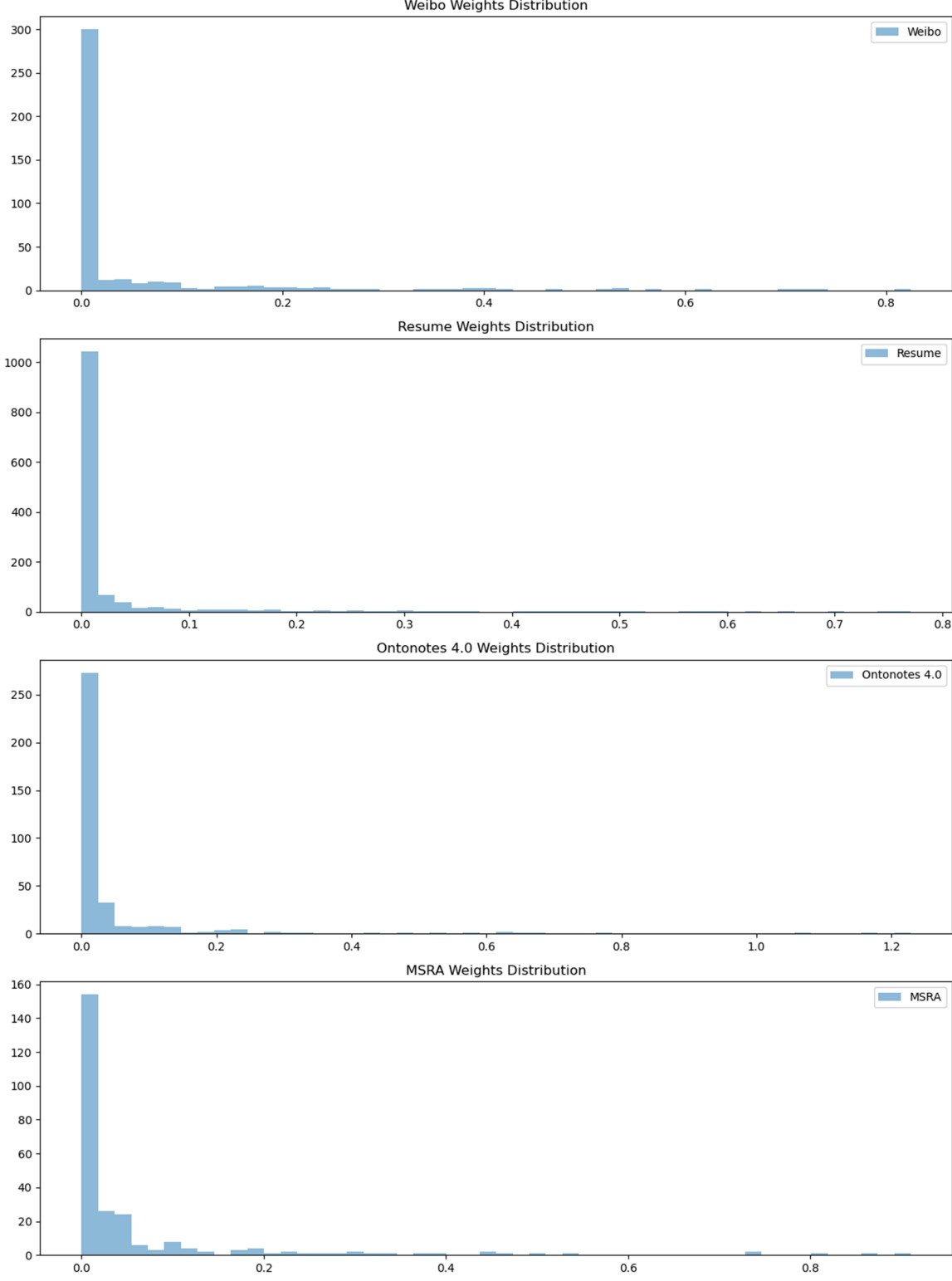

**Fig 10. Visualization of data weights distribution.**

"南京市长江大桥" (*Nanjing Yangtze River Bridge*), "市" (*City*) is labeled as 'E-Location' meaning the end of location entity type for word boundary "南京市" (*Nanjing City*) while labeled as 'B-Person' meaning the beginning of person entity type for word boundary "市长" (*Mayor*). So word boundaries be more informative with word types.

Without word boundaries, sentence structure cannot be effectively represented. Language is equal word and structure. From word boundaries, we can find the different levels of structure in sentences for syntactic and dependency analysis. It will help NER with the embedding of the graph structure. So, word boundaries are more informative with sentence structure.

## Theoretical upper bound of WBA improvement

The theoretical upper bound for improving NER using word boundary information is influenced by three types of errors: entity-type errors, word boundary errors, and word composition errors. In NER, common sources of error include incorrect entity boundaries and incorrect entity types. Since entities are typically single words or composed of multiple words, entity boundary errors arise from both word boundary errors and word composition errors. By incorporating accurate word boundary information, we can reduce entity boundary errors. Therefore, the theoretical upper bound for improvement is achieved when word boundary errors are completely eliminated. Let the entity type errors be denoted by $E_t$, the word boundary errors by $E_b$, and the word composition errors by $E_c$. Then: The theoretical upper bound of performance using word boundary information is when $E_b = 0$. The theoretical lower bound corresponds to the baseline case where no word boundary information is used. The potential improvement from using word boundary information is the difference between performance with $E_b = 0$ and the baseline.

## Clarity and theoretical explanation

We will provide a more comprehensive description of the theoretical contributions of WBA. Specifically, we will elaborate on the limitations of traditional attention mechanisms in NER, particularly for languages like Chinese that lack explicit word boundaries. The current attention-based models often struggle with segmenting tokens accurately, which leads to inefficiencies in recognizing named entities.

In contrast, WBA incorporates word boundaries explicitly into the attention mechanism, allowing the model to focus on segmenting words more effectively. This innovation is particularly important for Chinese, where the lack of clear word boundaries makes named entity recognition challenging.

To highlight the novelty of WBA, we will include a detailed comparison with existing models such as FLAT and NFLAT. While FLAT and NFLAT effectively address token-level attention in NER, they do not incorporate word boundaries in the attention process. This often leads to misclassifications in languages like Chinese. In comparison, WBA focuses on explicitly encoding word boundaries, which significantly improves accuracy in Chinese NER tasks. Additionally, we will include a visual comparison of the attention mechanisms used in NFLAT (word-char, Fig 11), FLAT (lattice-lattice, Fig 12) and WBA (char-char, Fig 8) to better illustrate the novel aspects of our approach. To illustrate the impact of WBA on word-level embeddings, we visualize the FLAT+WBA attention (lattice-lattice, Fig 13), simultaneously highlighting the role of word boundaries.

FLAT [37] and NFLAT [52] leverage lattice-based or non-lattice Transformers and explore various tokenization strategies. While these models improve upon previous methods, a significant limitation lies in their reliance on word lexicons for segmentation, often without fully accounting for the broader sentence context. This oversight can lead to incorrect token incorporations. For example, in the sentence "视频科比乔丹经典动作对比谁才是终结者分享自新浪视频" (*Video: Kobe vs. Jordan classic moves comparison — who is the real finisher? Shared from Sina Video.*), the character "作" (*Action*) receives attention weights of 1.00, 0.64, and 0.03 from "作对" (*Fight*) in NFLAT, FLAT, and FLAT+WBA, respectively. Both

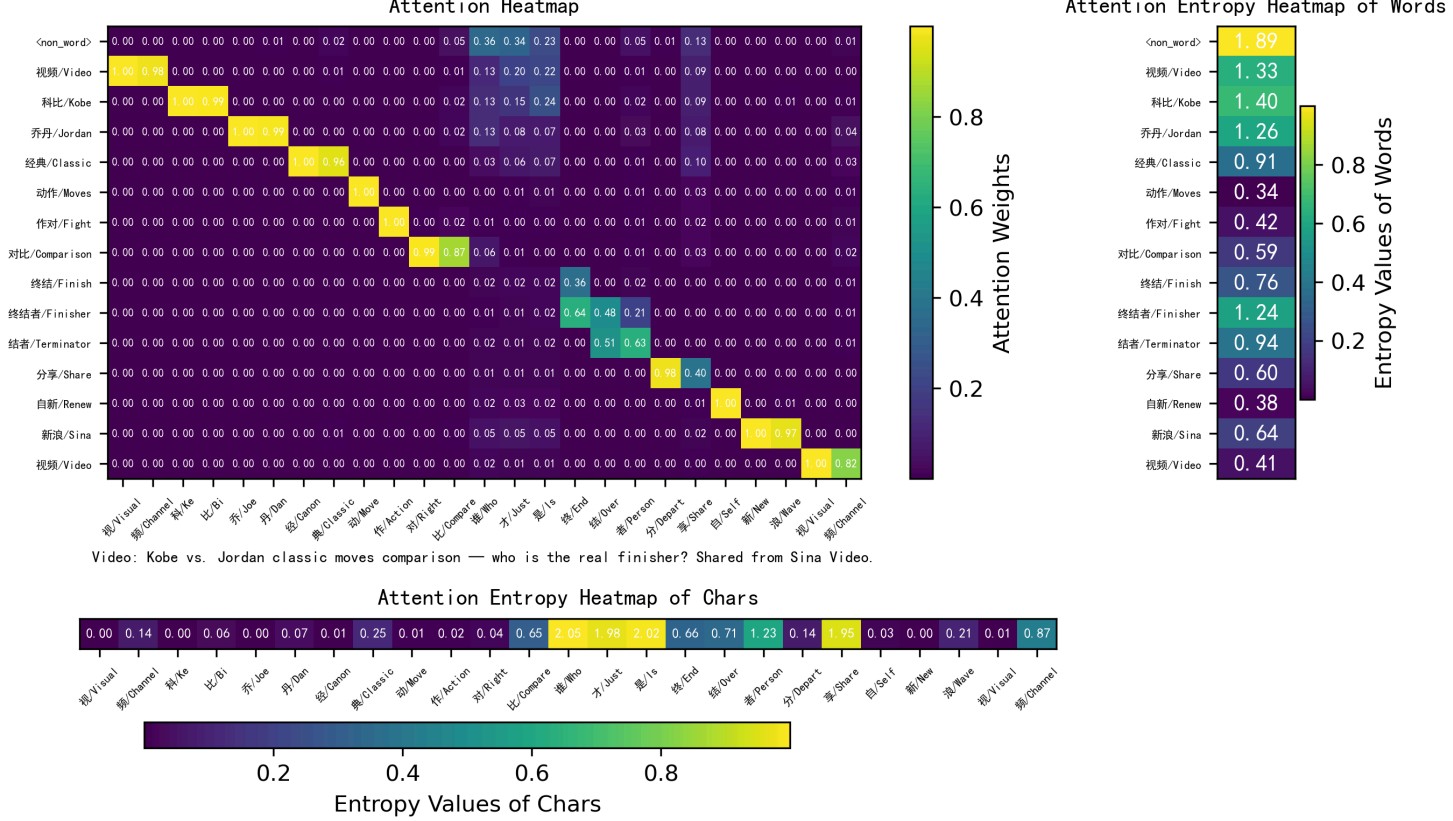

**Fig 11. Visualization of NFLAT attention.**

FLAT and NFLAT incorrectly treat "作对" (*Fight*) as a valid word, ignoring the wider sentence structure and resulting in erroneous segmentation.

In contrast, our method explicitly models word boundary attention across the entire sentence to mitigate word boundary errors, rather than relying solely on word lexicons. This approach provides a more accurate representation of sentence structure and better captures the relationships between nearby characters.

For Chinese NER, the self-attention mechanism in Transformers often suffers from sparsity and imbalance across characters. To overcome this, we introduce word-relative position attention, which softly identifies word boundaries while preserving contextual information. This method facilitates structural learning from large datasets, and, with the addition of boundary attention, the model can learn sentence structures more efficiently and effectively.

## Conclusion

We introduce a novel word boundary attention module, WBA, designed to focus on clear word boundaries in Chinese NER by identifying word points. WBA enhances character attention within their respective words, effectively bridging lexicon fusion and context encoding, leading to significant improvements in both performance and efficiency. Experimental results show that our method outperforms existing approaches, especially on the Weibo dataset, while maintaining

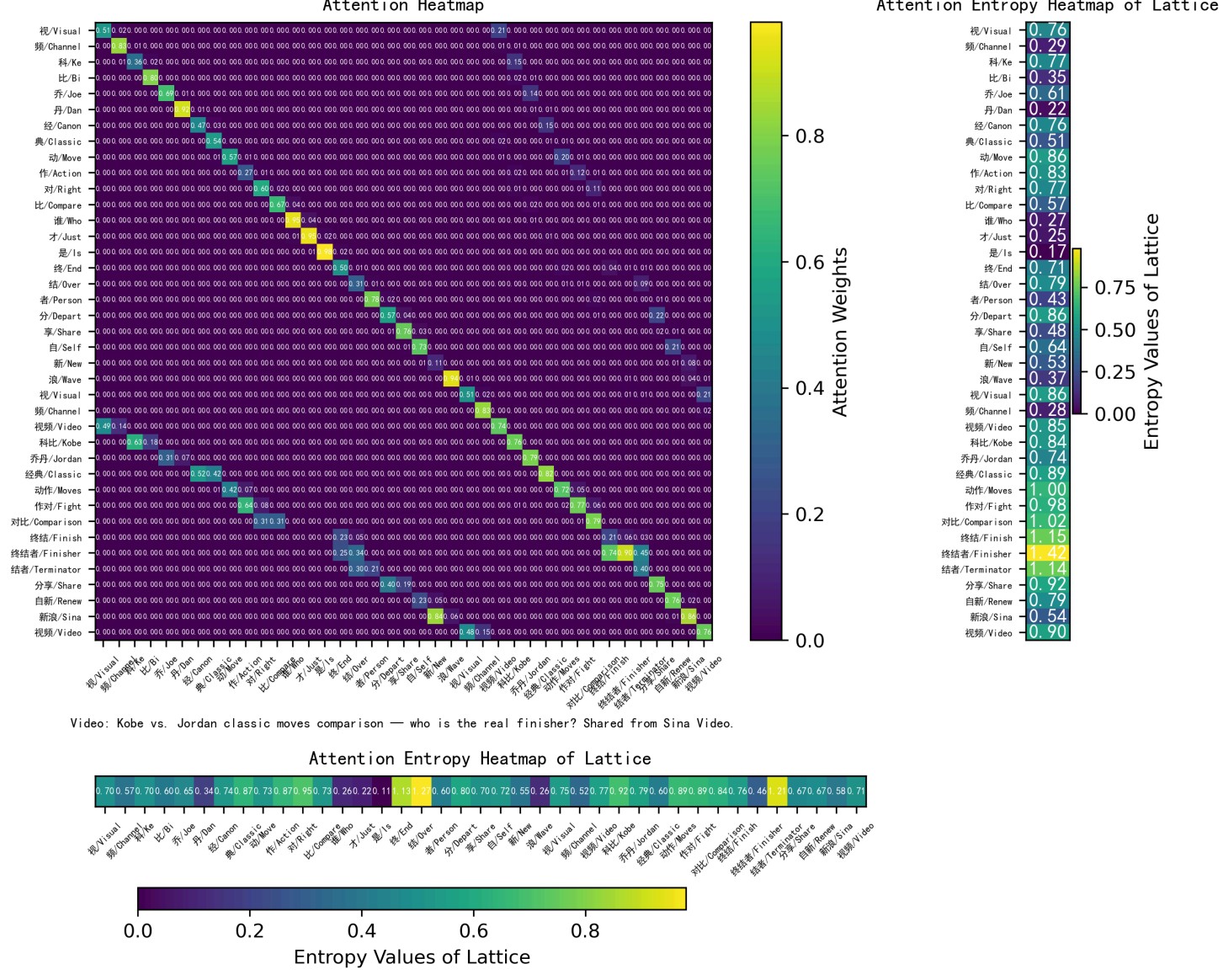

**Fig 12**. **Visualization of FLAT attention.**

simplicity and transferability. Moreover, WBA proves to be highly valuable for cross-dataset tasks, showcasing its versatility and potential for broader applications.

## Limitations

The NER method presented in this paper is specifically designed for Chinese. Future work could explore cross-lingual adaptations, potentially extending the approach to languages like Thai, which share similarities with Chinese in lacking clear word boundaries—unlike English, where spaces separate words. Additionally, the performance of our method is influenced by the quality, size, richness, and domain of the lexicon. These factors affect NER performance to varying degrees, with results varying based on different lexicon characteristics.

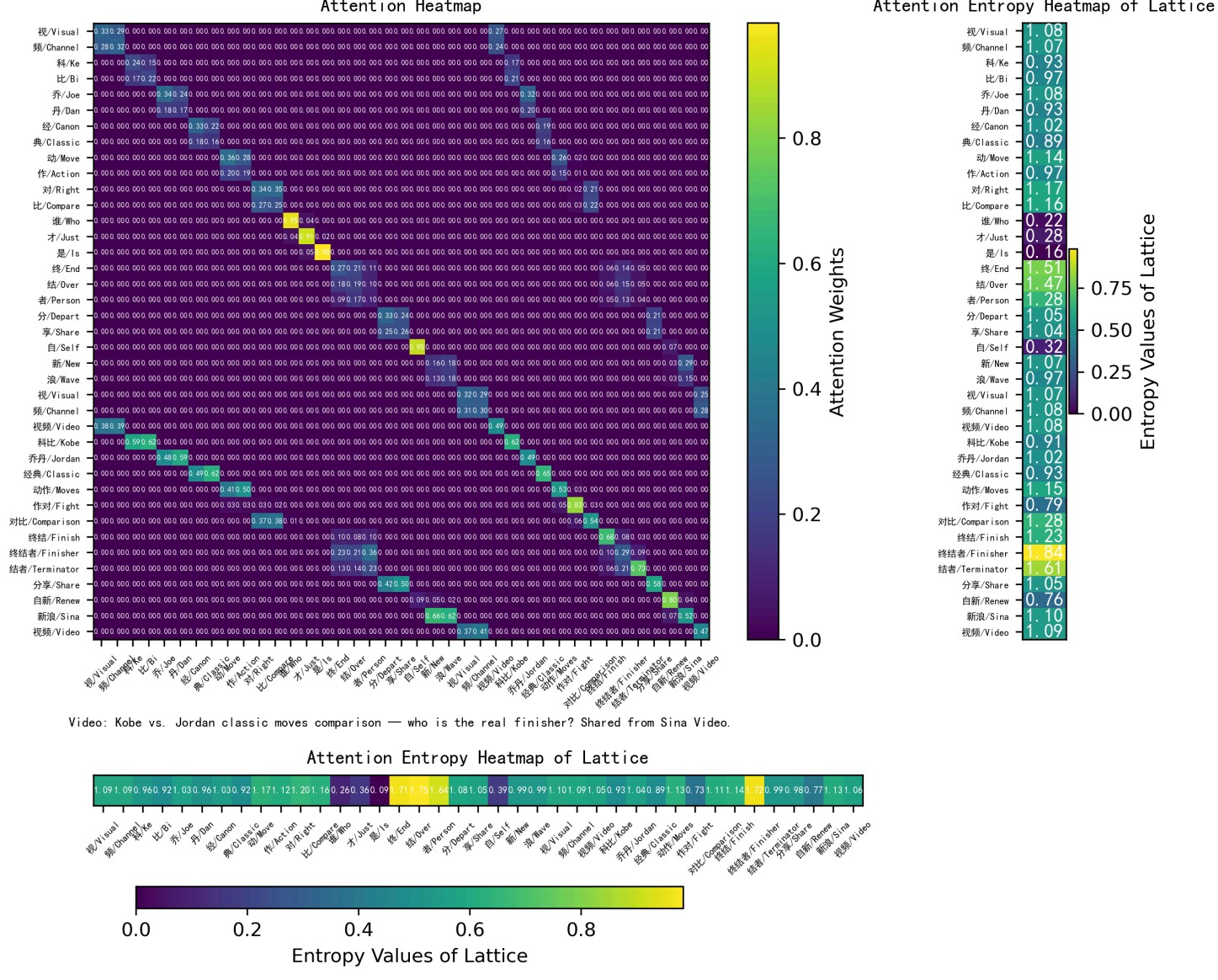

**Fig 13. Visualization of FLAT+WBA attention.**

## Acknowledgments

Thanks for editors and reviewers with comments to the authors.

## Author contributions

**Conceptualization:** Zhongguo Xu.

**Data curation:** Zhongguo Xu.

**Formal analysis:** Zhongguo Xu.

**Funding acquisition:** Zhongguo Xu.

**Investigation:** Zhongguo Xu.

**Methodology:** Zhongguo Xu.

**Project administration:** Zhongguo Xu.

**Resources:** Zhongguo Xu.

**Software:** Zhongguo Xu.

**Supervision:** Zhongguo Xu.

**Validation:** Zhongguo Xu.

**Visualization:** Zhongguo Xu.

**Writing – original draft:** Zhongguo Xu.

**Writing – review & editing:** Zhongguo Xu.

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
