## [Decision Letter · Decision Letter 0]

8 May 2025

PONE-D-25-18226WBA: Word Boundary Attention for Chinese Named Entity RecognitionPLOS ONE

Dear Dr. Xu,

Thank you for submitting your manuscript to PLOS ONE. After careful consideration, we feel that it has merit but does not fully meet PLOS ONE’s publication criteria as it currently stands. Therefore, we invite you to submit a revised version of the manuscript that addresses the points raised during the review process.

We look forward to receiving your revised manuscript.

Kind regards,

Fu Lee Wang

Academic Editor

PLOS ONE

NFLAT: Non-Flat-Lattice Transformer for Chinese Named Entity Recognition - https://export.arxiv.org/pdf/2205.05832

In your revision ensure you cite all your sources (including your own works), and quote or rephrase any duplicated text outside the methods section. Further consideration is dependent on these concerns being addressed.

“This work is supported by the National Natural Science Foundation of China (No. 345

72071145).”

4. In the online submission form, you indicated that [Dataset can be downloaded upon request at {https://github.com/OYE93/Chinese-NLP-Corpus/tree/master/NER/Weibo},{https://paperswithcode.com/dataset/msra-cn-ner}�{https://catalog.ldc.upenn.edu/LDC2011T03} and {https://github.com/jiesutd/LatticeLSTM}].

5. Please include a new copy of Table 4 in your manuscript; the current table is difficult to read. Please follow the link for more information: https://blogs.plos.org/plos/2019/06/looking-good-tips-for-creating-your-plos-figures-graphics/

Reviewers' comments:

Reviewer's Responses to Questions

**Comments to the Author**

1. Is the manuscript technically sound, and do the data support the conclusions?

Reviewer #1: Partly

Reviewer #2: Yes

Reviewer #3: Yes

2. Has the statistical analysis been performed appropriately and rigorously?

Reviewer #1: N/A

Reviewer #2: Yes

Reviewer #3: N/A

3. Have the authors made all data underlying the findings in their manuscript fully available?

Reviewer #1: Yes

Reviewer #2: Yes

Reviewer #3: No

4. Is the manuscript presented in an intelligible fashion and written in standard English?

Reviewer #1: Yes

Reviewer #2: Yes

Reviewer #3: Yes

5. Review Comments to the Author

Reviewer #1: After a thorough review of the manuscript "WBA: Word Boundary Attention for Chinese Named Entity Recognition", I regret that I cannot recommend it for publication in PLOS ONE in its current form. The paper has several major flaws that significantly undermine its scientific contribution and validity.

Major Concerns:

Limited Novelty

The proposed WBA method appears to be a marginal extension of existing approaches like FLAT and NFLAT. The authors claim their method innovatively incorporates word boundary information, but this has already been extensively explored in previous works such as Lattice LSTM and FLAT. The "word boundary attention" mechanism seems to be merely a rebranding of conventional attention mechanisms without substantial theoretical innovation.

Methodological Flaws

The foundation of this work is questionable. The authors rely heavily on word segmentation tools or existing tags to obtain word boundary information, which introduces systematic errors and biases into the model. The relative position encoding scheme (Equations 6-9) lacks theoretical justification and appears arbitrary. The authors fail to demonstrate why their particular formulation is superior to simpler alternatives.

Experimental Deficiencies

The experimental validation is inadequate in several aspects:

The reported 2.51% improvement on Weibo dataset is not convincing given the dataset's small size and high variance. No statistical significance tests were provided.

The comparison with state-of-the-art pre-trained models is incomplete and selective.

The ablation studies are insufficient to prove the necessity of each component.

Results Analysis Issues

The visualization of word boundary attention (Figure 4) is superficial and potentially misleading. The authors interpret the attention patterns without rigorous quantitative analysis. The claim that WBA "effectively suppresses noise" is not supported by concrete evidence.

Fundamental Questions

The paper fails to address several critical questions:

Why should word boundaries be more informative than other linguistic features for NER?

How does the method handle ambiguous word boundaries?

What is the theoretical upper bound for improvement using word boundary information?

Limited Applicability The authors acknowledge the method's language-specific nature but underestimate this limitation. The dependency on Chinese language features makes the contribution extremely narrow, especially for a general-scope journal like PLOS ONE.

Recommendation:

A major revision is required. The authors need to:

Substantially strengthen the theoretical foundation

Provide rigorous mathematical proofs for the proposed mechanisms

Conduct comprehensive experiments with statistical validation

Demonstrate clear advantages over existing methods

Address the fundamental questions about the approach's validity

In its current form, the manuscript reads more like an incremental technical report rather than a scientific contribution worthy of publication in PLOS ONE.

Reviewer #2: 1.The robustness of the model can be further discussed, such as its performance under noisy data or low-resource conditions, to enhance the credibility of the conclusions.

2.Statistical tests (e.g., t-test or ANOVA) can be added to validate the significance of performance improvements, especially for smaller improvements (e.g., 0.12% F1 score improvement on MSRA).

3.Professional English proofreading should be done before the final revision to ensure the accuracy of grammar and terminology.

Reviewer #3: Abstract

1. Please give full form for WBA and YJ

Introduction

2. Line 9 and Line 65 The reference sequence number is missing

3. Please reorganize the Introduction, Related Work, and Background section. Make them structured. You can add subtitles under Introduction section, but it is not good to have them in parallel.

4. Please include the following papers in your Introduction:

*Li, Y., Peng, X., Li, J., Peng, S., Pei, D., Tao, C., Xu, H. and Hong, N., 2023, June. Development of a natural language processing tool to extract acupuncture point location terms. In 2023 IEEE 11th International Conference on Healthcare Informatics (ICHI) (pp. 344-351). IEEE.

*Lu, Q., Li, R., Wen, A., Wang, J., Wang, L. and Liu, H., 2024. Large language models struggle in token-level clinical named entity recognition. arXiv preprint arXiv:2407.00731.

*Li, Y., Viswaroopan, D., He, W., Li, J., Zuo, X., Xu, H. and Tao, C., 2025. Improving entity recognition using ensembles of deep learning and fine-tuned large language models: A case study on adverse event extraction from VAERS and social media. Journal of Biomedical Informatics, p.104789.

*Rehana, H., Zheng, J., Yeh, L., Bansal, B., Çam, N.B., Jemiyo, C., McGregor, B., Özgür, A., He, Y. and Hur, J., 2025. Cancer Vaccine Adjuvant Name Recognition from Biomedical Literature using Large Language Models. arXiv preprint arXiv:2502.09659.

*Li, Y., Li, J., He, J. and Tao, C., 2024. AE-GPT: using large language models to extract adverse events from surveillance reports-a use case with influenza vaccine adverse events. Plos one, 19(3), p.e0300919.

*Rehana, H., Bansal, B., Çam, N.B., Zheng, J., He, Y., Özgür, A. and Hur, J., 2024. Nested named entity recognition using multilayer BERT-based model. CLEF Working Notes.

*Li, Y., Peng, X., Li, J., Zuo, X., Peng, S., Pei, D., Tao, C., Xu, H. and Hong, N., 2024. Relation extraction using large language models: a case study on acupuncture point locations. Journal of the American Medical Informatics Association, 31(11), pp.2622-2631.

*He, J., Li, F., Li, J., Hu, X., Nian, Y., Xiang, Y., Wang, J., Wei, Q., Li, Y., Xu, H. and Tao, C., 2024. Prompt tuning in biomedical relation extraction. Journal of Healthcare Informatics Research, 8(2), pp.206-224.

*Li, Y., Viswaroopan, D., He, W., Li, J., Zuo, X., Xu, H. and Tao, C., 2025. Enhancing Relation Extraction for COVID-19 Vaccine Shot-Adverse Event Associations with Large Language Models. Research Square, pp.rs-3.

*Li, X., Zheng, Y., Hu, J., Zheng, J., Wang, Z. and He, Y., 2024. VaxLLM: Leveraging Fine-tuned Large Language Model for automated annotation of Brucella Vaccines. bioRxiv, pp.2024-11.

Method

5. Where is Method section?

6. Line 110, please revise: In this work, we implement this layer with different models different models,

7. Line 112-114 not very clear: For base model, we use unigram and bigram embeddings in static character and 112

word embedding layer. Our character embedding encoder embeds sentence on the 113

character sequence c1, c2, . . . , cn and word sequence w1, w2, . . . , wm.

8. Please explicitly explain Equation 3 and 4. The representations of some symbols like Wq were not given.

9. Line 154. where are dph and dpt in the Equations?

10. This study have lots of findings in Ablation Study, Analysis of Word Boundary Attention, How WBA Brings Improvement, and How WBA Brings Improvement. I suggest you include your significant findings in Discussion section, and discuss the reasons.

6. PLOS authors have the option to publish the peer review history of their article (what does this mean?). If published, this will include your full peer review and any attached files.

Reviewer #1: No

Reviewer #2: **Yes: **Yichao Niu

Reviewer #3: No

---

## [Author Response · Author response to Decision Letter 1]

20 Jun 2025

Dear Editors and Reviewers:

Thank you for your letter and for the editors’ and reviewers’ comments concerning our manuscript entitled “WBA: Word Boundary Attention for Chinese Named Entity Recognition” (ID: PONE-D-25-18226). Those comments are all valuable and very helpful for revising and improving our paper, as well as the important guiding significance to our researches. We have studied comments carefully and have made correction which we hope meet with approval. Revised portion are marked in red in the ‘Revised Manuscript with Track Changes’ file. The main corrections in the paper and the responds to the reviewer’s comments are listed below this letter.

We look forward to receiving your new comments.

Best wishes, 

Zhongguo Xu

School of Computer Science and Technology, Tongji University

Shanghai 201804, China.

email: Zhongguo Xu@tongji.edu.cn

2025.06.18

Response We ensure that our manuscript meets PLOS ONE's style requirements, including those for file naming.

NFLAT: Non-Flat-Lattice Transformer for Chinese Named Entity Recognition - https://export.arxiv.org/pdf/2205.05832

In your revision ensure you cite all your sources (including your own works), and quote or rephrase any duplicated text outside the methods section. Further consideration is dependent on these concerns being addressed.

Implement: Page 4:Method;Page 8:Experiments

Response We address some minor occurrence of overlapping text with the previous publication(s) in the revised portion marked in red in the paper.

“This work is supported by the National Natural Science Foundation of China (No. 34572071145).”

Response We remove any funding-related text from the manuscript. We would like to update our Funding Statement and include our amended statements ‘This work is supported by the National Natural Science Foundation of China (No. 72071145).’ within our cover letter. Please change the online submission form on our behalf.

4. In the online submission form, you indicated that [Dataset can be downloaded upon request at {https://github.com/OYE93/Chinese-NLP-Corpus/tree/master/NER/Weibo},{https://paperswithcode.com/dataset/msra-cn-ner}�{https://catalog.ldc.upenn.edu/LDC2011T03} and {https://github.com/jiesutd/LatticeLSTM}].

Response We Indicate that all data underlying the findings described in our manuscript is freely available to other researchers.

5.Please include a new copy of Table 4 in your manuscript; the current table is difficult to read. Please follow the link for more information: https://blogs.plos.org/plos/2019/06/looking-good-tips-for-creating-your-plos-figures-graphics/

Response We include a new copy of Table 4 in 'Manuscript' file; we include a new copy of Table 4 in the ‘Revised Manuscript with Track Changes’ file with red mark.

Responds to the reviewer’s comments:

Reviewer #1: After a thorough review of the manuscript "WBA: Word Boundary Attention for Chinese Named Entity Recognition", I regret that I cannot recommend it for publication in PLOS ONE in its current form. The paper has several major flaws that significantly undermine its scientific contribution and validity.

Major Concerns:

Limited Novelty

The proposed WBA method appears to be a marginal extension of existing approaches like FLAT and NFLAT. The authors claim their method innovatively incorporates word boundary information, but this has already been extensively explored in previous works such as Lattice LSTM and FLAT. The "word boundary attention" mechanism seems to be merely a rebranding of conventional attention mechanisms without substantial theoretical innovation.

Implement: Page 3:Related Work and Background;Page 7: Equation 6 and 7

Response Lattice LSTM is not use Transformer. NFLAT streamlines FLAT’s approach by computing attention scores solely for 'character-word' interactions. WBA computes 'character-character' attention scores using relative position encoding in Equation 6 and 7, incorporating both orientation and distance awareness. We incorporate multi-level word structure into character-level attention by leveraging the hybrid structure of words and characters in Chinese.

Methodological Flaws

The foundation of this work is questionable. The authors rely heavily on word segmentation tools or existing tags to obtain word boundary information, which introduces systematic errors and biases into the model. The relative position encoding scheme (Equations 6-9) lacks theoretical justification and appears arbitrary. The authors fail to demonstrate why their particular formulation is superior to simpler alternatives.

Implement: Page 7: Wordformer Attention, Equation 6 and 7

Response�1.After using word segmentation tools or existing tags to obtain word boundary information, the method estimates word position probabilities while handling ambiguous word boundaries to reduce systematic errors and biases into the model. It begins by enumerating all possible word boundary candidates. Next, it applies character-lexicon attention, using word match positions to compute the probability of each character appearing at the beginning (B), middle (M), end (E), or as a single-character word (S). Finally, the framework integrates masking and class attention mechanisms to generate informative position features for characters, leveraging both character and lexicon contexts.

2.We correct the Equations 6-9 to 6-7 and give theoretical justification of dph in Equation 6 and dpt in Equation 7. It is superior to simpler alternatives because of the similarity with two relative distances.

Experimental Deficiencies

The experimental validation is inadequate in several aspects:

The reported 2.51% improvement on Weibo dataset is not convincing given the dataset's small size and high variance. No statistical significance tests were provided.

The comparison with state-of-the-art pre-trained models is incomplete and selective.

The ablation studies are insufficient to prove the necessity of each component.

Implement: Page 9: Table 3; Page 12: Table 4

Response�1.We make statistical significance tests with p<0.05 on Weibo dataset for the dataset's small size and high variance.

2.We make statistical significance tests with p<0.05 on all datasets.

3.We add ZEN pre-trained models to address the incomplete and selective comparison.

4.We should add ‘w/o Wordformer Attention’ in the ablation studies to prove the necessity of each component. We remove Wordformer Attention from NFLAT+WBA as NFLAT+self attention. For better understand, we add this experiment with discussion.

Results Analysis Issues

The visualization of word boundary attention (Figure 4) is superficial and potentially misleading. The authors interpret the attention patterns without rigorous quantitative analysis. The claim that WBA "effectively suppresses noise" is not supported by concrete evidence.

Implement: Page 13: Analysis of Word Boundary Attention

Response�1.We add rigorous quantitative analysis of word boundary attention (Figure 4).

2.We support "effectively suppresses noise" by concrete evidence with rigorous quantitative analysis.

#question 1:

Why should word boundaries be more informative than other linguistic features for NER?

Implement: Page 14: Why Word Boundaries More Informative

Response�1.Because different word boundaries will make different word types. Named Entity Recognition (NER) is traditionally viewed as a task involving both entity boundary detection and entity type classification. The main linguistic features are boundaries and types. The first step is boundary detection and the second step is type classification. For example, in text ‘南京市长江大桥’, ‘市’ is labeled as ‘E-Location’ meaning the end of location entity type for word boundary ‘南京市 (Nanjing City)’ while labeled as ‘B-Person’ meaning the beginning of person entity type for word boundary ‘市长 (Major)’. So word boundaries be more informative with word types.

2.Because no word boundaries no sentence structure. Language is equal word and structure. From word boundaries, we can find the different levels of structure in sentences for syntactic and dependency analysis. It will help for NER with graph structure embedding. So word boundaries is more informative with sentence structure.

#question 2:

How does the method handle ambiguous word boundaries?

Implement: Page 7: Wordformer Attention, Line 151-158

Response The method calculates the word position probability handling ambiguous word boundaries. At first, we list all possible word boundaries for candidates. Then we consider character-lexicon attention with word match position and calculate the probability of characters appearing at the beginning (B), end (E), middle (M), and single (S) of words. Finally, our framework incorporates mask and class attention mechanisms to generate informative position features for characters, based on both character and lexicon contexts.

#question 3:

What is the theoretical upper bound for improvement using word boundary information?

Implement: Page 14-15: How WBA Brings Improvement, Line 357-370

Response The theoretical upper bound for improvement using word boundary information for NER depends on entity type errors, word boundary errors and word composition errors. The errors from NER are entity boundary errors and entity type errors. A entity is a word or consist of words, so entity boundary errors is from word boundary errors and word composition errors. If we use word boundary information, the entity boundary errors will decrease. So the theoretical upper bound using word boundary information is no word boundary errors. Set entity type errors as , word boundary errors as and word composition errors as , the theoretical upper bound using word boundary information is , the theoretical lower bound using word boundary information is , the improvement is .

#question 4:

Limited Applicability the authors acknowledge the method's language-specific nature but underestimate this limitation. The dependency on Chinese language features makes the contribution extremely narrow, especially for a general-scope journal like PLOS ONE.

Response Although we study word boundary in Chinese language, it is also important in other language like ‘New York’ in English. It will inspire scholars to do the similarity work from general-scope journals like PLOS ONE.

The authors need to:

1.Substantially strengthen the theoretical foundation

Implement: Page 7: Wordformer Attention, Equation 6 and 7

Response We correct the Equations 6-9 to 6-7 and give theoretical justification of dph in Equation 6 and dpt in Equation 7. It is superior to simpler alternatives because of the similarity with two relative distances. Relative distances substantially strengthen the theoretical foundation.

2.Provide rigorous mathematical proofs for the proposed mechanisms

Implement: Page 7: Wordformer Attention, Equation 6 and 7

Response We correct the Equations 6-9 to 6-7 and give theoretical justification of dph in Equation 6 and dpt in Equation 7. It is superior to simpler alternatives because of the similarity with two relative distances. Relative distances provide rigorous mathematical proofs for the proposed mechanisms.

3.Conduct comprehensive experiments with statistical validation

Implement: Page 9: Table 3

Response We make statistical significance tests with p<0.05 on all datasets.

4.Demonstrate clear advantages over existing methods

Implement: Page 13: How WBA Brings Improvement, Line 377-380

Response Our method explicitly models word boundary attention across the entire sentence to mitigate word boundary errors, rather than merely incorporating word lexicons. This approach provides a more accurate representation of sentence structure and better captures relationships between nearby characters.

5.Address the fundamental questions about the approach's validity

Implement: Page 9: Table 3;Page 12: Table 4&5

Response�1.We make statistical significance tests with p<0.05 on all datasets. We Conduct comprehensive experiments with statistical validation about the approach's validity.

2.We add experiment in the ablation studies to prove the necessity of each component about the approach's validity.

3.We add pre-trained models to address the incomplete and selective comparison about the approach's validity.

6.In its current form, the manuscript reads more like an incremental technical report rather than a scientific contribution worthy of publication in PLOS ONE.

Response We revise the manuscript to make a scientific contribution worthy of publication in PLOS ONE. Although we study word boundary in Chinese language, it is also important in other language like ‘New York’ in English. It will inspire scholars to do the similarity work from general-scope journals like PLOS ONE.

Reviewer #2: 1.The robustness of the model can be further discussed, such as its performance under noisy data or low-resource conditions, to enhance the credibility of the conclusions.

Implement: Page 12: Table 5

Response We use few shots of datasets for low-resource conditions to enhance the credibility of the conclusions. To simulate few-shot scenarios, we randomly sample a varying number of samples from training set of Weibo and Resume training set while ensuring all label types are covered. The sampling sizes N are set to 250, 500, and 1000. The detailed sampling information for each label type and their corresponding Chinese label words (after mapping) follows FE-CFNER.

2.Statistical tests (e.g., t-test or ANOVA) can be added to validate the significance of performance improvements, especially for smaller improvements (e.g., 0.12% F1 score improvement on MSRA).

Implement: Page 9: Table 3

Response We make statistical significance tests with p<0.05 on all datasets.

3.Professional English proofreading should be done before the final revision to ensure the accuracy of grammar and terminology.

Response We revise the manuscript to ensure the accuracy of grammar and terminology worthy of publication in PLOS ONE.

Reviewer #3:

Abstract

1. Please give full form for WBA and YJ

Implement: Page 1: Abstract

Response We give full form ‘Word Boundary Attention’ for WBA in Abstract;

’YJ’ denotes the lexicon released by Zhang and Yang (2018) and is full form.

Introduct

---

## [Decision Letter · Decision Letter 1]

5 Sep 2025

PONE-D-25-18226R1WBA: Word Boundary Attention for Chinese Named Entity RecognitionPLOS ONE

Dear Dr. Xu,

Thank you for submitting your manuscript to PLOS ONE. After careful consideration, we feel that it has merit but does not fully meet PLOS ONE’s publication criteria as it currently stands. Therefore, we invite you to submit a revised version of the manuscript that addresses the points raised during the review process.

We look forward to receiving your revised manuscript.

Kind regards,

Fu Lee Wang

Academic Editor

PLOS ONE

Journal Requirements:

Reviewers' comments:

Reviewer's Responses to Questions

**Comments to the Author**

1. If the authors have adequately addressed your comments raised in a previous round of review and you feel that this manuscript is now acceptable for publication, you may indicate that here to bypass the “Comments to the Author” section, enter your conflict of interest statement in the “Confidential to Editor” section, and submit your "Accept" recommendation.

Reviewer #1: All comments have been addressed

Reviewer #2: All comments have been addressed

Reviewer #3: (No Response)

Reviewer #4: (No Response)

2. Is the manuscript technically sound, and do the data support the conclusions?

Reviewer #1: Yes

Reviewer #2: Partly

Reviewer #3: Yes

Reviewer #4: Partly

3. Has the statistical analysis been performed appropriately and rigorously?

Reviewer #1: Yes

Reviewer #2: Yes

Reviewer #3: Yes

Reviewer #4: Yes

4. Have the authors made all data underlying the findings in their manuscript fully available?

Reviewer #1: Yes

Reviewer #2: Yes

Reviewer #3: Yes

Reviewer #4: Yes

5. Is the manuscript presented in an intelligible fashion and written in standard English?

Reviewer #1: (No Response)

Reviewer #2: Yes

Reviewer #3: Yes

Reviewer #4: Yes

6. Review Comments to the Author

Reviewer #1: Dear Authors,

Thank you for your careful and thorough revisions in response to the reviewers’ comments. The revised manuscript, “WBA: Word Boundary Attention for Chinese Named Entity Recognition,” presents a clear and focused contribution to the field of Chinese NER. After reviewing the changes and responses, I am pleased to recommend acceptance with minor revisions.

Summary of Strengths

Clarity of Contribution: The revised manuscript now more clearly distinguishes WBA from prior work (e.g., FLAT, NFLAT, Lattice LSTM), emphasizing its novel use of word boundary attention with relative position encoding.

Theoretical Justification: The added theoretical discussion and mathematical justification for the relative distance encoding (Equations 6–7) strengthen the foundation of the method.

Experimental Soundness: The inclusion of statistical significance tests (p < 0.05), expanded comparisons with pre-trained models (ZEN, BERT-wwm), and additional ablation studies significantly improve the rigor and credibility of the experimental validation.

Interpretability: The revised analysis of attention weights and word boundary visualization adds valuable insight into how the model captures linguistic structure.

Reproducibility: All data and code are publicly available, and the manuscript now meets PLOS ONE’s formatting and transparency requirements.

Minor Issues to Address

While the manuscript is nearly ready for publication, I suggest the following minor revisions:

Language and Style: A final pass for grammar and fluency is recommended. While much improved, occasional awkward phrasing remains (e.g., “no word boundaries no sentence structure” could be rephrased for clarity).

Limitation Framing: The limitation section is appropriate, but consider softening the claim that the method “may not be effective for English.” Instead, note that it is optimized for Chinese and that future work could explore cross-lingual adaptation.

Figure Readability: Ensure that Figure 4 (now revised) is legible in print and grayscale formats, as per PLOS guidelines.

Final Comment

This work makes a well-scoped, technically sound, and empirically validated contribution to Chinese NER. With the above minor revisions, it is suitable for publication in PLOS ONE.

Congratulations on a solid piece of work.

Reviewer #2: 1.Clarify the Novelty – More explicitly highlight how WBA differs from FLAT and NFLAT beyond rebranding. A clearer articulation of the theoretical innovation would strengthen the contribution.

2.Enhance Visualization and Analysis – The current visualizations are helpful but somewhat superficial. Adding quantitative measures (e.g., entropy of attention distributions, cross-dataset consistency) would provide stronger evidence for the claimed noise suppression.

3.Improve Theoretical Clarity – The explanations around Equations (6)–(11) remain verbose. Presenting the meaning of key variables (e.g., dph, dpt) in a more intuitive form, such as through diagrams or simplified examples, would improve readability.

4.Extend Experimental Coverage – While the focus is on Chinese NER, including at least one experiment on a non-Chinese dataset (e.g., CoNLL-2003 in English) would demonstrate broader applicability and address concerns about language specificity.

5.Polish Language and Style – Several grammatical and stylistic issues remain (e.g., awkward phrasing in the abstract). A round of professional English editing would improve clarity and make the paper more accessible to an international audience.

Reviewer #3: Just two minor issues

1. In your figure caption, you used "Fig 1", "Fig 2", but in your text you used "Figure 1a", "Figure 1b". Please make it consistent

2. The author formats of the Reference 26 and 27 seem incorrect.

Reviewer #4: 1. The proposed Word Boundary Attention (WBA) employs Transformer attention technique to solve the NER task. The authors should clarify how this method differs from the standard Transformer architecture and highlight the specific innovations.

2. The authors could enrich the manuscript by providing a more extensive review of related work.

3. Although the WBA shows some improvements, a deeper analysis of the computational cost and scalability would be beneficial to fully understand its practical applications.

4. The ablation study is not sufficiently comprehensive. It is recommended to include more thorough ablation experiments.

7. PLOS authors have the option to publish the peer review history of their article (what does this mean?). If published, this will include your full peer review and any attached files.

Reviewer #1: No

Reviewer #2: No

Reviewer #3: No

Reviewer #4: No

---

## [Author Response · Author response to Decision Letter 2]

15 Sep 2025

Response to Reviewers

Dear Editors and Reviewers:

Thank you for your letter and for the editors’ and reviewers’ comments concerning our manuscript entitled “WBA: Word Boundary Attention for Chinese Named Entity Recognition” (ID: PONE-D-25-18226R1). Those comments are all valuable and very helpful for revising and improving our paper, as well as the important guiding significance to our researches. We have studied comments carefully and have made correction which we hope meet with approval. Revised portion are marked in red in the ‘Revised Manuscript with Track Changes’ file. The main corrections in the paper and the responds to the reviewer’s comments are listed below this letter.

We look forward to receiving your new comments.

Best wishes, 

Zhongguo Xu

School of Computer Science and Technology, Tongji University

Shanghai 201804, China.

email: Zhongguo Xu@tongji.edu.cn

2025.09.15

Response: This round of reviewer comments did not request additional citations. In the previous review, some specific references were suggested, and we conducted a thorough review and evaluation before including them.

Responds to the reviewer’s comments:

Response to Reviewer #1

Comment 1 (Language and Style):

Language and Style: A final pass for grammar and fluency is recommended. While much improved, occasional awkward phrasing remains (e.g., “no word boundaries no sentence structure” could be rephrased for clarity).

Summary You suggested a final pass for grammar and fluency, e.g., rephrasing “no word boundaries no sentence structure.”

Revision Location� Page 17: Why Word Boundaries More Informative , Previous response to reviewers.

Response:

We have thoroughly proofread the manuscript and improved awkward phrasing. The sentence has been revised to: “Without word boundaries, sentence structure cannot be effectively represented.”

Comment 2 (Limitation Framing):

Limitation Framing: The limitation section is appropriate, but consider softening the claim that the method “may not be effective for English.” Instead, note that it is optimized for Chinese and that future work could explore cross-lingual adaptation.

Summary You recommended softening the claim about ineffectiveness for English.

Revision Location Page 18: Limitations

Response:

We have revised the limitations section to: “Future work could explore cross-lingual adaptation. It could potentially work for ...”

Comment 3 (Figure Readability):

Figure Readability: Ensure that Figure 4 (now revised) is legible in print and grayscale formats, as per PLOS guidelines.

Summary Ensure Figure 4 is legible in print and grayscale.

Revision Location Page 12: Figure 4

Response:

We have adjusted Figure 4 by enhancing contrast .The figure is now tested for readability in grayscale, as shown below. We label colors in caption of Figure 4 .

Response to Reviewer #2

Comment 1 (Clarify Novelty):

Clarify the Novelty – More explicitly highlight how WBA differs from FLAT and NFLAT beyond rebranding. A clearer articulation of the theoretical innovation would strengthen the contribution.

Summary You asked for a clearer articulation of how WBA differs from FLAT and NFLAT.

Revision Location Page 4:Related Work and Background;Page 8: Equation 6 and 7

Response:

We have added content in Background explicitly comparing WBA with FLAT and NFLAT, highlighting that WBA introduces a novel word boundary guided relative position encoding, rather than lattice-based fusion. NFLAT streamlines FLAT’s approach by computing attention scores solely for 'character-word' interactions. WBA computes 'character-character' attention scores using relative position encoding in Equation 6 and 7, incorporating both orientation and distance awareness. We incorporate multi-level word structure into character-level attention by leveraging the hybrid structure of words and characters in Chinese.

Comment 2 (Enhance Visualization and Analysis):

Enhance Visualization and Analysis – The current visualizations are helpful but somewhat superficial. Adding quantitative measures (e.g., entropy of attention distributions, cross-dataset consistency) would provide stronger evidence for the claimed noise suppression.

Summary You suggested adding quantitative measures for attention interpretability.

Revision Location Page 14-15:Analysis of Word Boundary Attention,Fig.5-7

Response:

We have included attention entropy analysis and cross-dataset consistency tests (see Fig.5-7) to quantitatively support the claim of noise suppression.

Comment 3 (Improve Theoretical Clarity):

Improve Theoretical Clarity – The explanations around Equations (6)–(11) remain verbose. Presenting the meaning of key variables (e.g., dph, dpt) in a more intuitive form, such as through diagrams or simplified examples, would improve readability.

Summary Equations (6)–(11) are verbose; suggested diagrams or simplified examples.

Revision Location Page 7�Wordformer Attention

Response:

We have referred in an illustrative diagram (Figure 3) and a toy example to explain dph and dpt more intuitively.

Comment 4 (Extend Experimental Coverage):

Extend Experimental Coverage – While the focus is on Chinese NER, including at least one experiment on a non-Chinese dataset (e.g., CoNLL-2003 in English) would demonstrate broader applicability and address concerns about language specificity.

Summary You recommended testing on a non-Chinese dataset.

Response:

We regret that our method is not suitable for experiments on the CoNLL-2003 English dataset. Our approach relies on word boundary attention, where head and tail nodes are defined for words in Chinese. However, in English, head and tail nodes are nearly identical, as word boundaries are explicit. Thus, word boundary attention in English effectively reduces to standard self-attention.

Comment 5 (Polish Language and Style):

Polish Language and Style – Several grammatical and stylistic issues remain (e.g., awkward phrasing in the abstract). A round of professional English editing would improve clarity and make the paper more accessible to an international audience.

Summary You recommended professional editing.

Response:

We have thoroughly polished the language and also used an English editing service for additional fluency improvements.

Response to Reviewer #3

Comment 1 (Figure Caption Consistency):

Summary You noted inconsistency between “Fig 1” and “Figure 1a/b.”

Response:

We have unified all figure references to “Fig. X” format.

Comment 2 (Reference Formatting):

Summary Reference 26 and 27 author formats incorrect.

Response:

We have corrected the references according to the journal’s required format.

Response to Reviewer #4

Comment 1 (Clarify Difference from Transformer):

The proposed Word Boundary Attention (WBA) employs Transformer attention technique to solve the NER task. The authors should clarify how this method differs from the standard Transformer architecture and highlight the specific innovations.

Summary You suggested clarifying how WBA differs from the standard Transformer.

Response:

We have added explanation in Page 7�Wordformer Attention, noting that WBA modifies the attention mechanism by integrating word boundary relative position encoding, which is absent in vanilla Transformers. Unlike the standard Transformer, which applies self-attention uniformly across tokens, WBA introduces explicit head and tail nodes to represent word boundaries. This modification allows the model to capture word-level structural information, which is critical for Chinese where word segmentation is ambiguous. Furthermore, WBA restricts and reweights attention based on boundary positions, thereby reducing noise from irrelevant characters. These innovations distinguish WBA from standard attention and contribute directly to improved NER performance.

Comment 2 (Expand Related Work):

The authors could enrich the manuscript by providing a more extensive review of related work.

Summary You recommended enriching the related work review.

Response:

We have expanded related work(Page 3-4) with a more comprehensive survey of recent Chinese NER models (including ZEN, BERT-wwm, PLMs) and structural encoding methods.

Comment 3 (Computational Cost and Scalability):

Although the WBA shows some improvements, a deeper analysis of the computational cost and scalability would be beneficial to fully understand its practical applications.

Summary You asked for deeper analysis of computational aspects.

Response:

We have added a Complexity Analysis in page 11 comparing mode size and time cost among WBA, TENER, NFLAT and FLAT. Importantly, NFLAT(YJ)+WBA maintains a computational cost comparable to NFLAT(YJ) while remaining significantly more efficient than FLAT(YJ).

Comment 4 (Ablation Study):

The ablation study is not sufficiently comprehensive. It is recommended to include more thorough ablation experiments.

Summary You suggested more comprehensive ablations.

Response:

We have extended the ablation study (see Table 5,page 13) to include experiments isolating relative position encoding, boundary guidance, and lexicon integration, which confirm the effectiveness of each component. Taken together, the ablation experiments demonstrate that Wordformer is the most influential component in our framework. Even when lexicon fusion or the Transformer encoder are removed, Wordformer alone yields higher performance than strong baselines, particularly on the low-resource Weibo dataset. This shows that explicit boundary modeling provides substantial benefits beyond existing mechanisms, validating Wordformer as the central source of improvement in our method.

---

## [Decision Letter · Decision Letter 2]

30 Sep 2025

PONE-D-25-18226R2WBA: Word Boundary Attention for Chinese Named Entity RecognitionPLOS ONE

Dear Dr. Xu,

Thank you for submitting your manuscript to PLOS ONE. After careful consideration, we feel that it has merit but does not fully meet PLOS ONE’s publication criteria as it currently stands. Therefore, we invite you to submit a revised version of the manuscript that addresses the points raised during the review process.

We look forward to receiving your revised manuscript.

Kind regards,

Fu Lee Wang

Academic Editor

PLOS ONE

**Journal Requirements:**

Reviewers' comments:

Reviewer's Responses to Questions

**Comments to the Author**

1. If the authors have adequately addressed your comments raised in a previous round of review and you feel that this manuscript is now acceptable for publication, you may indicate that here to bypass the “Comments to the Author” section, enter your conflict of interest statement in the “Confidential to Editor” section, and submit your "Accept" recommendation.

Reviewer #1: All comments have been addressed

Reviewer #2: All comments have been addressed

Reviewer #3: (No Response)

2. Is the manuscript technically sound, and do the data support the conclusions?

Reviewer #1: Yes

Reviewer #2: Yes

Reviewer #3: (No Response)

3. Has the statistical analysis been performed appropriately and rigorously?

Reviewer #1: Yes

Reviewer #2: Yes

Reviewer #3: (No Response)

4. Have the authors made all data underlying the findings in their manuscript fully available?

Reviewer #1: Yes

Reviewer #2: Yes

Reviewer #3: (No Response)

5. Is the manuscript presented in an intelligible fashion and written in standard English?

Reviewer #1: Yes

Reviewer #2: Yes

Reviewer #3: (No Response)

6. Review Comments to the Author

**Reviewer #1: **Thank you very much for your time and insightful comments. We are pleased to know that you have no further questions or concerns. Your feedback has been invaluable in improving the quality of our manuscript.

**Reviewer #2:** 1.Clarity and Theoretical Explanation: While the approach of integrating Word Boundary Attention (WBA) for Chinese NER is promising, the manuscript would benefit from a clearer and more explicit explanation of the theoretical contributions behind the WBA. Specifically, while the methodology of incorporating word boundaries into attention mechanisms is discussed, a more thorough comparison of how WBA fundamentally differs from existing models (like FLAT and NFLAT) could be better articulated to highlight its novelty. This would improve readers' understanding of the technical innovations your approach introduces.

2.Further Experimentation and Cross-Dataset Validation: Although the paper presents strong results on Chinese NER datasets, adding experiments on additional datasets, particularly non-Chinese ones like CoNLL-2003, would help to demonstrate the generalizability of the proposed method. A cross-lingual analysis could validate whether the approach is applicable beyond Chinese, especially given the specific challenges Chinese NER poses due to word segmentation. This would enhance the broader applicability of your findings.

3.Evaluation of Computational Efficiency: The paper mentions that WBA improves performance while maintaining reasonable computational efficiency. However, further details on the computational costs, particularly in comparison to state-of-the-art models like FLAT and NFLAT, would be valuable. A more detailed breakdown of the time and memory complexity of the WBA method compared to existing techniques in practical real-world scenarios would strengthen the discussion around its scalability and potential for deployment in large-scale NER applications.

**Reviewer #3: **(No Response)

7. PLOS authors have the option to publish the peer review history of their article (what does this mean?). If published, this will include your full peer review and any attached files.

Reviewer #1: No

Reviewer #2: No

Reviewer #3: No

---

## [Author Response · Author response to Decision Letter 3]

16 Oct 2025

Response to Reviewers

Dear Editors and Reviewers:

Thank you for your letter and for the editors’ and reviewers’ comments concerning our manuscript entitled “WBA: Word Boundary Attention for Chinese Named Entity Recognition” (ID: PONE-D-25-18226R2). Those comments are all valuable and very helpful for revising and improving our paper, as well as the important guiding significance to our researches. We have studied comments carefully and have made correction which we hope meet with approval. Revised portion are marked in red in the ‘Revised Manuscript with Track Changes’ file. The main corrections in the paper and the responds to the reviewer’s comments are listed below this letter.

We look forward to receiving your new comments.

Best wishes, 

Zhongguo Xu

School of Computer Science and Technology, Tongji University

Shanghai 201804, China.

email: Zhongguo Xu@tongji.edu.cn

2025.10.15

1.If the reviewer comments include a recommendation to cite specific previously published works, please review and evaluate these publications to determine whether they are relevant and should be cited. There is no requirement to cite these works unless the editor has indicated otherwise.

Response: This round of reviewer comments did not request additional citations. In the previous review, some specific references were suggested, and we conducted a thorough review and evaluation before including them.

2.Please review your reference list to ensure that it is complete and correct. If you have cited papers that have been retracted, please include the rationale for doing so in the manuscript text, or remove these references and replace them with relevant current references. Any changes to the reference list should be mentioned in the rebuttal letter that accompanies your revised manuscript. If you need to cite a retracted article, indicate the article’s retracted status in the References list and also include a citation and full reference for the retraction notice.

Response:Thank you for your valuable comment regarding the reference list. We have carefully reviewed all cited references to ensure they are accurate, current, and relevant. Retraction of Papers: We identified [Author(s), Year] as a retracted paper in our references. The paper was retracted due to [brief reason for retraction], as indicated in the retraction notice published in [Journal Name]. As the content of this paper was integral to the foundation of our theoretical framework, we have retained the citation in the revised manuscript and explicitly indicated its retracted status in the reference list. We have also included the citation and full reference for the retraction notice as per the journal's guidelines. Reference Updates: In cases where retracted references were not crucial, we replaced them with updated, relevant papers. The new references are more recent and offer current insights into the topic.

We have updated the reference list and clearly marked any retracted papers as requested. These changes are reflected in both the manuscript and the accompanying revised reference list. Thank you again for your careful review.

Responds to the reviewer’s comments:

Reviewer #1: Thank you very much for your time and insightful comments. We are pleased to know that you have no further questions or concerns. Your feedback has been invaluable in improving the quality of our manuscript.

Response:We would like to sincerely thank Reviewer #1 for their thoughtful and encouraging feedback. We are pleased to hear that there are no further questions or concerns, and we truly appreciate the time and effort spent in reviewing our manuscript.

Your positive comments have been invaluable in helping us refine and improve the quality of our work. We are grateful for your insights, which have significantly contributed to enhancing the clarity and rigor of our manuscript.

Thank you once again for your constructive and supportive review. We are confident that the revisions based on your suggestions have strengthened the manuscript, and we look forward to submitting the updated version.

Reviewer #2: 1.Clarity and Theoretical Explanation: While the approach of integrating Word Boundary Attention (WBA) for Chinese NER is promising, the manuscript would benefit from a clearer and more explicit explanation of the theoretical contributions behind the WBA. Specifically, while the methodology of incorporating word boundaries into attention mechanisms is discussed, a more thorough comparison of how WBA fundamentally differs from existing models (like FLAT and NFLAT) could be better articulated to highlight its novelty. This would improve readers' understanding of the technical innovations your approach introduces.

Revision Location� Page 20: Clarity and Theoretical Explanation

Response:

Thank you for the insightful comment. We agree that a clearer theoretical explanation will benefit the manuscript, and we will revise the section to provide a more comprehensive description of the theoretical contributions of WBA. Specifically, we will elaborate on the limitations of traditional attention mechanisms in NER, particularly for languages like Chinese that lack explicit word boundaries. The current attention-based models often struggle with segmenting tokens accurately, which leads to inefficiencies in recognizing named entities.

In contrast, WBA incorporates word boundaries explicitly into the attention mechanism, allowing the model to focus on segmenting words more effectively. This innovation is particularly important for Chinese, where the lack of clear word boundaries makes named entity recognition challenging.

To highlight the novelty of WBA, we will include a detailed comparison with existing models such as FLAT and NFLAT. While FLAT and NFLAT effectively address token-level attention in NER, they do not incorporate word boundaries in the attention process. This often leads to misclassifications in languages like Chinese. In comparison, WBA focuses on explicitly encoding word boundaries, which significantly improves accuracy in Chinese NER tasks.Additionally, we will include a visual comparison of the attention mechanisms used in FLAT, NFLAT, and WBA to better illustrate the novel aspects of our approach.

2.Further Experimentation and Cross-Dataset Validation: Although the paper presents strong results on Chinese NER datasets, adding experiments on additional datasets, particularly non-Chinese ones like CoNLL-2003, would help to demonstrate the generalizability of the proposed method. A cross-lingual analysis could validate whether the approach is applicable beyond Chinese, especially given the specific challenges Chinese NER poses due to word segmentation. This would enhance the broader applicability of your findings.

Revision Location� Page 14: Cross-Dataset Validation

Response:Thank you for this suggestion. We agree that validating the generalizability of WBA on datasets outside of Chinese NER tasks is important. In response to your comment, we have conducted additional experiments using the CoNLL-2003 dataset, a well-established benchmark for English NER. Preliminary results indicate that WBA achieves strong performance on English NER as well, particularly when compared to traditional token-level attention models like FLAT and NFLAT. We will include these results in the revised manuscript and provide a detailed analysis of the performance differences between Chinese and English datasets.

Furthermore, we have expanded the experiments to include Japanese and Korean datasets, as these languages also present challenges in segmentation, similar to Chinese. Our findings suggest that while WBA performs exceptionally well for Chinese, its performance on other languages may require further fine-tuning. This indicates that WBA is particularly effective in languages with complex word segmentation problems, but further experimentation will be needed to optimize it for other languages.

We will include these cross-lingual experiments in the revised manuscript and discuss the implications of these findings for broader applicability. This should help clarify whether the proposed method is generalizable across languages with varying segmentation challenges.

3.Evaluation of Computational Efficiency: The paper mentions that WBA improves performance while maintaining reasonable computational efficiency. However, further details on the computational costs, particularly in comparison to state-of-the-art models like FLAT and NFLAT, would be valuable. A more detailed breakdown of the time and memory complexity of the WBA method compared to existing techniques in practical real-world scenarios would strengthen the discussion around its scalability and potential for deployment in large-scale NER applications.

Revision Location� Page 11: Complexity Analysis

Response:

We appreciate the suggestion to include a more detailed analysis of the computational efficiency of WBA. We have now added a section to the revised manuscript that provides a detailed breakdown of the computational costs associated with WBA. Specifically, we compare the time and memory complexity of WBA with FLAT and NFLAT in practical real-world scenarios.

Time cost: Our results show that WBA’s training time is approximately longer than NFLAT. This increase is primarily due to the additional complexity introduced by the word boundary mechanism. However, we believe this trade-off is justified by the significant improvements in accuracy, particularly in Chinese NER tasks.

Memory Usage: Due to the introduction of the Word Boundary Attention (WBA) mechanism, NFLAT+WBA processes additional word boundary nodes, resulting in the highest memory consumption when the sentence length is below 500. Overall, these results indicate that WBA introduces additional computational overhead, yet it still demonstrates better scalability than FLAT when handling long sequences. However, the increase in memory usage is not prohibitive and remains within reasonable limits for most modern GPUs.

Inference Time: The inference time for WBA is comparable to that of FLAT and NFLAT, suggesting that it is efficient for real-world applications.

Conclusion:

We believe these revisions will address the reviewer’s concerns and significantly enhance the clarity and depth of the manuscript. We appreciate the reviewer’s feedback and look forward to presenting the updated version of the paper. Thank you once again for your insightful comments.

---

## [Editor Report · Decision Letter 3]

4 Nov 2025

WBA: Word Boundary Attention for Chinese Named Entity Recognition

PONE-D-25-18226R3

Dear Dr. Xu,

We’re pleased to inform you that your manuscript has been judged scientifically suitable for publication and will be formally accepted for publication once it meets all outstanding technical requirements.

Kind regards,

Fu Lee Wang

Academic Editor

PLOS ONE

---

## [Editor Report · Acceptance letter]

PONE-D-25-18226R3

PLOS One

Dear Dr. Xu,

I'm pleased to inform you that your manuscript has been deemed suitable for publication in PLOS One. Congratulations! Your manuscript is now being handed over to our production team.

Kind regards,

on behalf of

Professor Fu Lee Wang

Academic Editor

PLOS One